# Dynamic SARS-CoV-2 emergence algorithm for rationally-designed logical next-generation vaccines

David P. Maison[1,2,3], Lauren L. Ching[1,2,3], Sean B. Cleveland[4,5], Alanna C. Tseng[1,2,3], Eileen Nakano[1,2,3], Cecilia M. Shikuma[1,3,6] & Vivek R. Nerurkar[1,2,3 ✉]

SARS-CoV-2 worldwide spread and evolution has resulted in variants containing mutations resulting in immune evasive epitopes that decrease vaccine efficacy. We acquired SARS-CoV-2 positive clinical samples and compared the worldwide emerged spike mutations from Variants of Concern/Interest, and developed an algorithm for monitoring the evolution of SARS-CoV-2 in the context of vaccines and monoclonal antibodies. The algorithm partitions logarithmic-transformed prevalence data monthly and Pearson's correlation determines exponential emergence of amino acid substitutions (AAS) and lineages. The SARS-CoV-2 genome evaluation indicated 49 mutations, with 44 resulting in AAS. Nine of the ten most worldwide prevalent (>70%) spike protein changes have Pearson's coefficient $r > 0.9$. The tenth, D614G, has a prevalence >99% and $r$-value of 0.67. The resulting algorithm is based on the patterns these ten substitutions elucidated. The strong positive correlation of the emerged spike protein changes and algorithmic predictive value can be harnessed in designing vaccines with relevant immunogenic epitopes. Monitoring, next-generation vaccine design, and mAb clinical efficacy must keep up with SARS-CoV-2 evolution, as the virus is predicted to remain endemic.

[1] Department of Tropical Medicine, Medical Microbiology, and Pharmacology, University of Hawai'i at Mānoa, Honolulu, HI, USA. [2] Pacific Center for Emerging Infectious Diseases Research, University of Hawai'i at Mānoa, Honolulu, HI, USA. [3] John A. Burns School of Medicine, University of Hawai'i at Mānoa, Honolulu, HI, USA. [4] Hawaii Data Science Institute, University of Hawai'i at Mānoa, Honolulu, HI, USA. [5] Department of Cyberinfrastructure, University of Hawai'i at Mānoa, Honolulu, HI, USA. [6] Hawaii Center for AIDS, University of Hawai'i at Mānoa, Honolulu, HI, USA. ✉email: nerurkar@hawaii.edu

Since the origin of the Coronavirus Disease 2019 (COVID-19) pandemic, severe acute respiratory syndrome coronavirus-2 (SARS-CoV-2) has rapidly evolved into seven Variants of Interest (VOI) and four Variants of Concern (VOC)[1]. Further, as of July 15, 2021, over 2,559,000 SARS-CoV-2 genomic sequences have been deposited in the publicly available GenBank and the Global Initiative on Sharing Avian Influenza Data (GISAID) databases[2]. From the establishment of the now universal D614G substitution[3] to the emergence of the VOC and VOI with dozens of different mutations across their respective genomes[1], SARS-CoV-2 evolution has been evident throughout the pandemic. To define these evolutionary events, the Centers for Disease Control and Prevention (CDC) has classified certain lineages as VOC and VOI to denote highly adapted and immunologically evasive strains of SARS-CoV-2 based on expert evaluation of available data by the SARS-CoV-2 Interagency Group (SIG)[1]. More recently, the World Health Organization (WHO) has further given its own classification to emerging SARS-CoV-2 lineages using letters of the Greek alphabets[4].

Hawai'i has been disproportionately affected by COVID-19 in terms of race, wherein 20% of the cases occur in 4% of the population of Pacific Islanders[5,6]. Understanding the SARS-CoV-2 as it relates to emerging lineages in Hawai'i, an isolated island community with diverse ethnic groups, will allow for a greater understanding of the pandemic's nature worldwide.

Fortunately, early in the pandemic, governments and private sectors around the globe poured resources into producing efficacious vaccines. In the United States, three of these vaccines (Pfizer and BioNtech BNT162b2, Moderna mRNA-1273, and Janssen Ad26.COV2.S) are authorized and recommended by the U.S. Food and Drug Administration (FDA)[7,8]. Unfortunately, all these vaccines have reduced efficacy against all VOC[1], an effect likely to amplify further as the virus evolves from the vaccine design of the original strain. The loss of efficacy can be attributed to the alteration of immunogenic epitopes[9]. These epitope alterations are also shown to diminish monoclonal antibody effectiveness[10–12]. Several of these mutations are found in the spike protein used in vaccine design, and therefore allows the virus to evade antibodies targeted to the original strain of that vaccine. Similar to the annual influenza virus vaccine, the evolution of SARS-CoV-2 presents the dilemma of how to redesign next-generation vaccines to keep up with the evolution of the virus.

One attempt to match the vaccine to the evolution of the virus was by Moderna. In response to the B.1.351 VOC considerably reducing the efficacy of the Novavax vaccine[13], Moderna explored the use of the B.1.351 VOC sequence in their mRNA vaccine design[14]. Promisingly, the newly adapted vaccine increased neutralization against the B.1.351 VOC when given as a booster[15]. However, the B.1.351 was only 1.15% prevalent worldwide in April 2021. Therefore, the continuous clinical trial evaluation against emerging VOC is not practical to match the rate and diversity with which mutations and VOC emerge worldwide.

To answer the dilemma of redesigning next-generation SARS-CoV-2 vaccines and predicting monoclonal antibody effectiveness in populations, we present and further validate our previously described quantitative analysis[16] for determining the emergence of individual substitutions and deletions and variants, alike, as an algorithm. This algorithm is a platform for monitoring the virus and determining appropriate vaccine design as we proceed into the evolution and endemicity of SARS-CoV-2[17]. Additionally, adapting the vaccines to match the evolution of the viral sequences may help alleviate the increased morbidity and mortality among the diverse ethnic groups in Hawai'i[5,6]. We utilized SARS-CoV-2 isolated in Hawai'i, and whole genome sequences (WGS) deposited in GenBank and GISAID, in combination with

VOC and VOI to validate the algorithm, a prototype alpha test for rationally-designing logical next-generation vaccines. As SARS-CoV-2 evolves, so must SARS-CoV-2 monitoring and vaccine design.

## Results

**Virus isolation, growth kinetics, and genomic equivalence.** Virus was isolated from an oropharyngeal (OPS) collected five days following symptom onset from an individual with PCR confirmed SARS-CoV-2 infection and propagated in Vero E6 cells. A stock of the SARS-CoV-2, isolate USA-HI498 2020 was produced following three passages in Vero E6 cells. Minimal CPE was observed at 12 h, moderate CPE at 24 h, and remarkable CPE at 48 h (Fig. 1a–d). Plaque assays were used to quantitate SARS-CoV-2 isolates. The virus stocks were titered at $1.28 \times 10^7$ PFU/mL for SARS-CoV-2 isolate USA-HI498 and $3.88 \times 10^7$ PFU/mL for SARS-CoV-2 USA-WA1/2020. Viral copy number analysis using N1, N2 and RdRp primers as well as microscopic observation showed no differences between the SARS-CoV-2, isolate USA-HI498 2020 and SARS-CoV-2 USA-WA1/2020 (Fig. 1e–m). SARS-CoV-2, isolate USA-HI498 2020 is deposited in BEI Resources Cat# NR-56130 and the corresponding whole genome sequence in the GenBank (Accession Number MZ664037).

**Whole genome sequencing and informatics.** RNA extraction was confirmed using RT-PCR[16]. There were 702,978 and 792,952 reads for SARS-CoV-2, isolate USA-HI498 2020 and HI708, respectively (Supplementary Table 1). FastQC confirmed the quality of both the untrimmed and trimmed fastqc files. There was no change in the nucleotide sequences between the VTM and the stock virus. The whole genome sequences for both the original VTM (OK021552, OK189251) and the stock virus (MZ664037, MZ664038) are deposited in the GenBank.

**Hawai'i SARS-Cov-2 sequences and lineage searches.** From GenBank, 317 full-genome SARS-CoV-2 sequences were obtained on July 28, 2021. Further, an additional 2942 sequences were obtained from GISAID. Hawai'i has 52 unique lineages in the 3259 representative sequences (A.1, A.2.2, A.3, AY.1, AY.2, B, B.1, B.1.1, B.1.1.207, B.1.1.222, B.1.1.304, B.1.1.316, B.1.1.380, B.1.1.416, B.1.1.519, B.1.1.7, B.1.108, B.1.139, B.1.160, B.1.2, B.1.234, B.1.241, B.1.243, B.1.265, B.1.298, B.1.340, B.1.351, B.1.357, B.1.36.8, B.1.369, B.1.37, B.1.400, B.1.413, B.1.427, B.1.429, B.1.517, B.1.526, B.1.561, B.1.568, B.1.575, B.1.588, B.1.595, B.1.596, B.1.601, B.1.609, B.1.617.2, B.1.623, B.6, P.1, P.1.1, P.2, R.1). As of April 12, 2021, GISAID reported 8809 sequences of B.1.243 lineage worldwide. The B.1.243 variant was represented by 717 of the 1002 (72%) sequences curated from Hawai'i. This prevalence decreased to 23% by July 28, 2021. Worldwide, GISAID reported 8,809 sequences of B.1.243 lineage.

**Quantitation of SARS-Cov-2 variants and amino acid substitution/deletions in comparison to epitope mapping of the spike protein.** The output S gene alignment between the thirteen genomic sequences (12 variants and one reference Wuhan sequence) identified 49 single-nucleotide polymorphisms (SNP). Pearson's correlation on logarithmically-transformed prevalence was calculated for the 12 SARS-CoV-2 variants in this study in order of highest to lowest *r* value, and separately for the omicron variants, as outlined in Supplementary Table 2, Figs. 2, 3 and and Supplementary Fig. 2.

Further, of the 49 identified SNP in the S gene, 44 resulted in non-synonymous AAS and deletions in the protein, and 5 were synonymous as outlined in Figs. 4a, b, c.i-xii and Supplementary Tables 2 and 3. Pearson's correlation on logarithmically-transformed

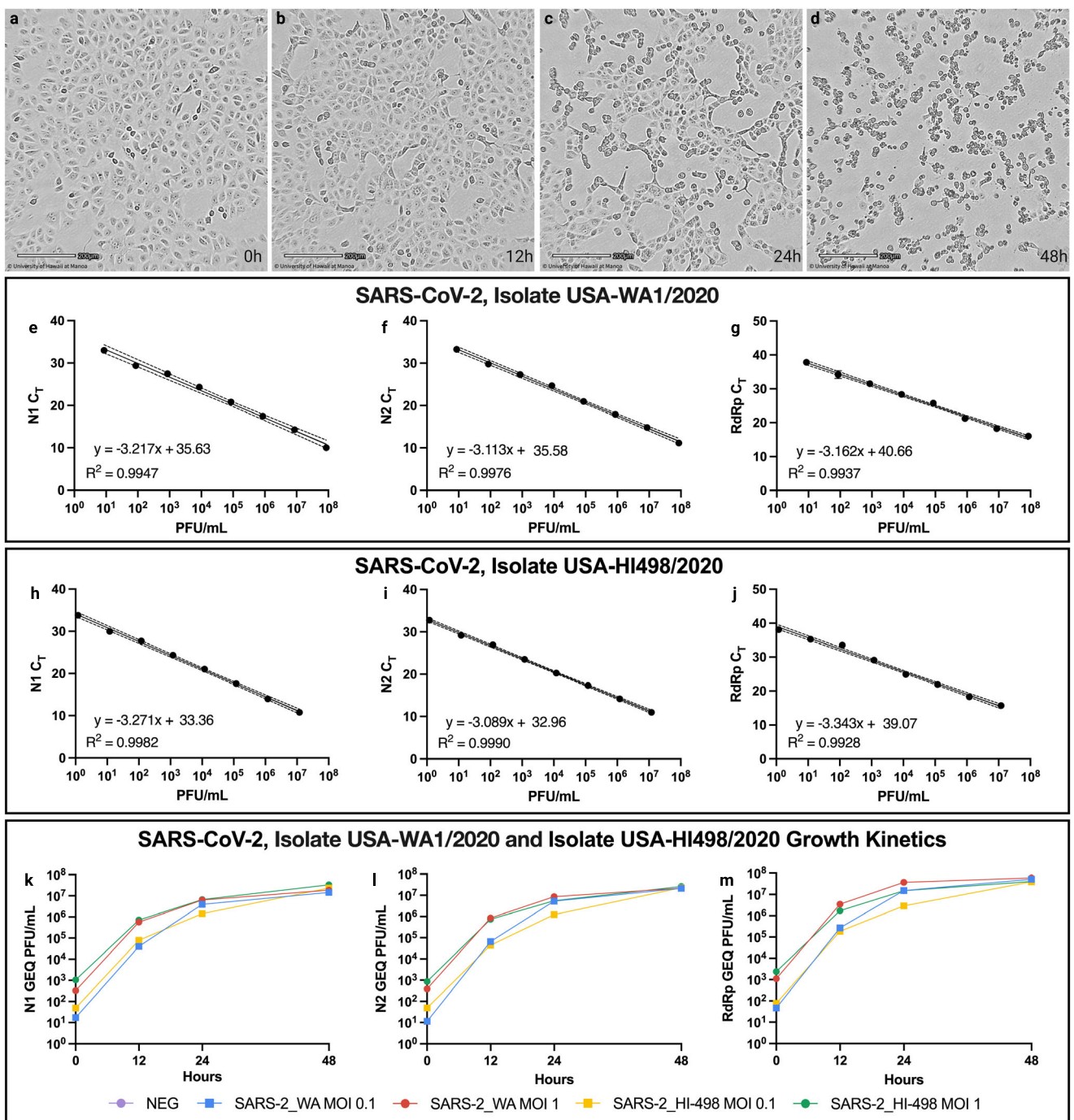

**Fig. 1 Cytopathic Effect and Growth Kinetics of SARS-CoV-2, Isolate USA-HI498 2020.** The figure shows time-lapse cell images of VeroE6 cells infected with SARS-CoV-2 isolate USA-HI498 2020 at different time points, demonstrating the cytopathic effect (CPE) induced by the virus at multiplicity of infection (MOI) 1. **a** 0 h, **b** 12 h, **c** 24 h, and **d** 48 h. Scale bar equals 200 μm. **e** Genomic equivalent (GEQ) of SARS-CoV-2, isolate USA-WA-1/2020 with N1 primers. **f** GEQ of SARS-CoV-2, isolate USA-WA-1/2020 with N2 primers. **g** GEQ of SARS-CoV-2, isolate USA-WA-1/2020 with RdRp primers. **h** GEQ of SARS-CoV-2, isolate USA-HI498/2020 with N1 primers. **i** GEQ of SARS-CoV-2, isolate USA-HI498/2020 with N2 primers. **j** GEQ of SARS-CoV-2, isolate USA-HI498/2020 with RdRp primers. **k** Growth kinetics of SARS-CoV-2 isolates USA-HI498/2020 and USA-WA-1/2020 at multiplicity of infection (MOI) 1 and 0.1 over 48 h using the N1 GEQ (e and h). **l** Growth kinetics of SARS-CoV-2 isolates USA-HI498/2020 (yellow and green) and USA-WA-1/2020 (blue and red) at MOI 1 and 0.1 over 48 h using the N2 GEQ (**f** and **i**). **m** Growth kinetics of SARS-CoV-2 isolates USA-HI498/2020 and USA-WA-1/2020 at MOI 1 and 0.1 over 48 h using the RdRp GEQ (**g** and **j**).

prevalence was calculated for all identified SARS-CoV-2 AAS and deletions (Supplementary Tables 2, 3, Fig. 5 and Supplementary Fig. 1).

The PubMed search for epitope predictions returned 42 publications. In total, 393 in silico predicted B and T cell epitopes corresponding to the spike protein were mapped from these publications (Fig. 4b). Of these, 108 epitopes involved the N-terminal domain (NTD), 102 epitopes involved the receptor binding domain (RBD), 7 epitopes involved the S1/S2 furin cleavage site, 10 epitopes involved the fusion peptide, 20 involved

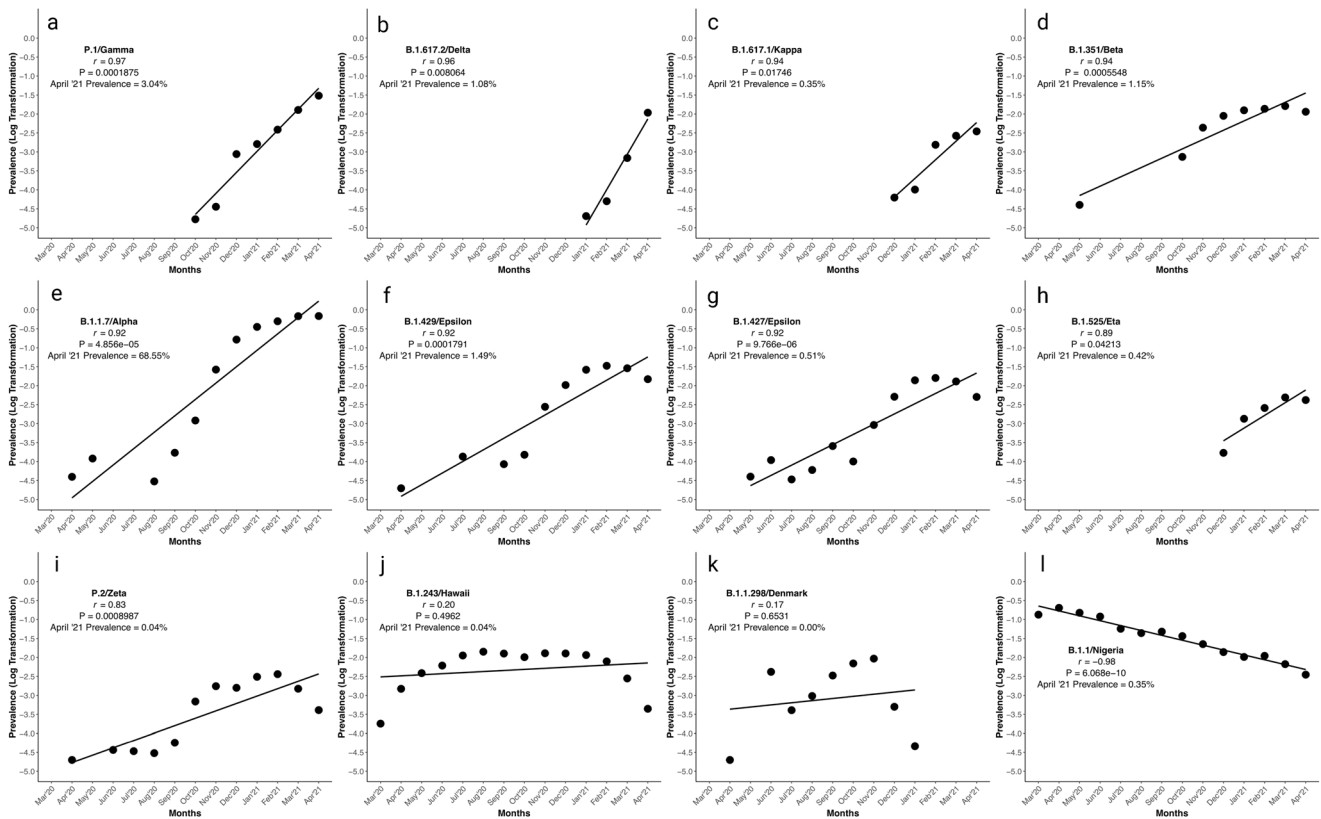

**Fig. 2 Pearson's correlation on logarithmically-transformed prevalence ratios of SARS-CoV-2 variant of concern, variants of interest, and other lineages.** This figure demonstrates the quantitation of SARS-CoV-2 variants of concern, variants of interest, and other lineages. The emergence and disappearance of variants/lineages of SARS-CoV-2 is evaluated by Pearson's correlation of logarithmic transformation prevalence data ($n = 1,479,378$ biologically independent samples). Variants are displayed in order of decreasing $r$ value. Pearson's correlation of logarithmic transformed prevalence versus time as a interval value for SARS-CoV-2 lineages **a** P.1, **b** B.1.617.2, **c** B.1.617.1, **d** B.1.351, **e** B.1.1.7, **f** B.1.429, **g** B.1.427, **h** B.1.525, **i** P.2, **j** B.1.243, **k** B.1.1.298, and **l** B.1.1. Graphs were generated using open-source RStudio version 1.3.1093 (R version 4.0.3) and the ggplot2 package under MIT + license (https://cran.r-project.org/web/packages/ggplot2/index.html). Graphs were compiled and the final figure created using Biorender.com.

heptad repeat 1, 12 involved heptad repeat 2, 12 involved the transmembrane region, and 8 involved the intracellular tail domain. The remaining 112 epitopes fell outside of these domains. Further, 239 and 151 epitopes were in the S1 and S2, respectively, with at least one predicted epitope covering 97% of the spike protein.

**Variant comparison.** To demonstrate the implementation of the algorithm, in Supplementary Table 3, we have shown the unique nucleotide mutations and resulting AAS and deletions for each of the twelve SARS-CoV-2 variants, in comparison to the reference sequence. BNT162b2 (Pfizer) and mRNA-1273 (Moderna) vaccines both contain two AAS (K986P and V987P) (Fig. 4c.xii)[18]. Novavax and Janssen vaccines also contain two AAS, K986P and V987P. Further, additional AAS at the furin cleavage site includes, R862Q, R683Q, and R685Q (Novavax), and R682S and R685G (Janssen) (Fig. 4c.xiii)[18].

**B.1.243 phylogeny and origin tracking.** The Hawai'i SARS-CoV-2 sequences in the GenBank and GISAID were combined with all worldwide B.1.243 lineages to produce an initial MAFFT alignment of 8820 sequences. Further, 4273 sequences with ambiguities and 1596 duplicate sequences were removed. The final alignment for constructing the phylogenetic tree was 2953 unique and unambiguous B.1.243 sequences. Using this method[19], we were able to define the origin of SARS-CoV-2, isolate USA-HI498 2020 and HI708 (Fig. 6).

**Discussion**

In this report, we lay the foundation for an adaptive and rational algorithm for monitoring SARS-CoV-2 evolution and quantitating variants in the context of vaccine design. Further, we describe the isolation, genetic characterization, phylogenetic analysis, and immunogenetic epitopes of the spike protein based on the SARS-CoV-2 lineage B.1.243 from Hawai'i. We employed B.1.243 to establish and validate the algorithm, as well as analyze VOC and VOI in the context of emerging spike protein amino acid changes for surveillance and future vaccine design.

Hawai'i has not been spared from this pandemic, and diverse ethnic populations here are disproportionately infected with SARS-CoV-2 as compared to other races and ethnicities[5]. Following isolation and identification of the B.1.243 lineage from the isolate SARS-CoV-2, isolate USA-HI498 2020, and virus strain HI708, we found through curation and analysis of published sequences that the B.1.243 lineage was once the dominating lineage in Hawai'i, causing more than 40% of all cases. The B.1.243 lineage that once dominated in overall prevalence in Hawai'i was introduced from Washington, California, Pennsylvania, and New Mexico, with the majority of sequences arising from horizontal transfer within Hawai'i. SARS-CoV-2, isolate USA-HI498 2020 was introduced to Hawai'i from New Mexico and the HI-708 SARS-CoV-2 strain originated from California. As this report is written, similar to the continental United States the SARS-CoV-2 Delta variant is rapidly spreading in Hawai'i[20].

At the time of the submission, the CDC identified four SARS-CoV-2 variants as VOC: B.1.1.7 (Alpha), B.1.351 (Beta), B.1.617.2

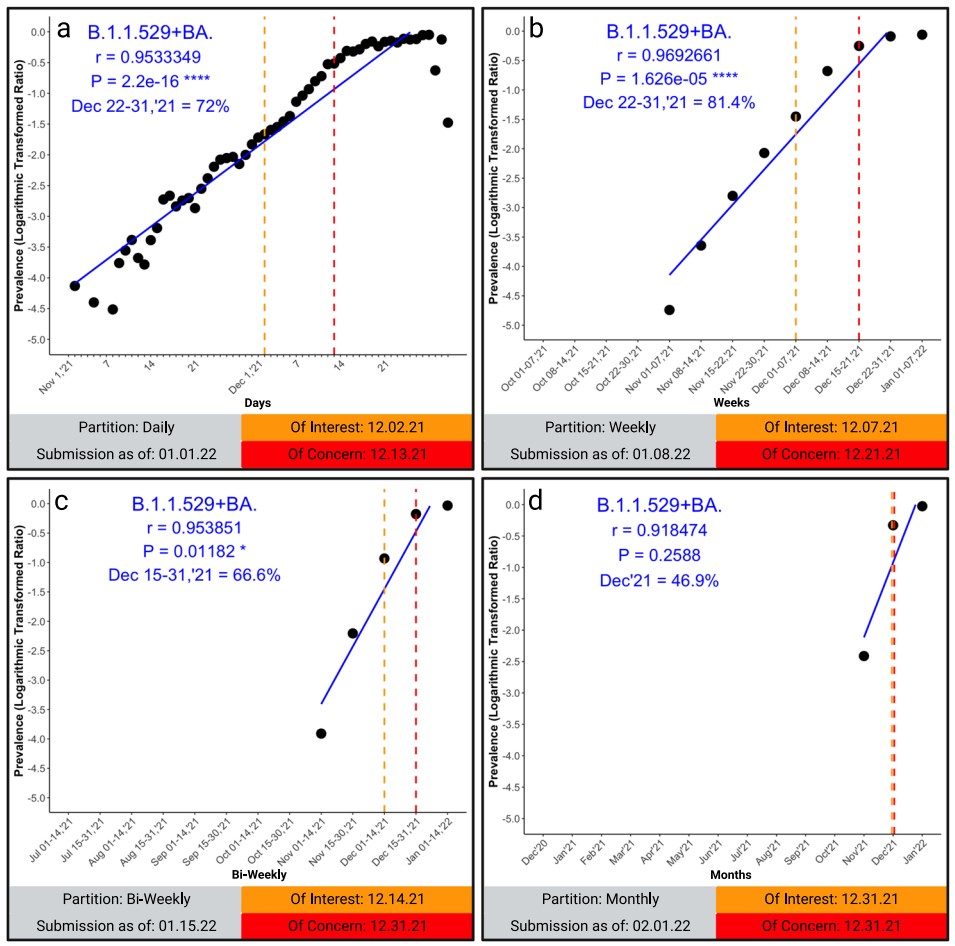

**Fig. 3 Pearson's correlation on logarithmically-transformed prevalence ratios of the omicron SARS-CoV-2 variant of concern.** This figure demonstrates the quantitation of the Omicron SARS-CoV-2 variant of concern. The emergence is shown for the collective variant (B.1.1.529+BA.*). The partitions are demonstrated in daily partitions (**a**) ($n = 1,077,671$ biologically independent samples), weekly partitions (**b**) ($n = 1,375,929$ biologically independent samples), bi-weekly partitions (**c**) ($n = 1,582,268$ biologically independent samples), and the prototype monthly partition (**d**) ($n = 2,154,954$ biologically independent samples). Each lineage and partition displays the date the VOC would be classified "of interest" (orange/orange dashed-lines) and "of concern" (red/red dashed-lines) as defined by the Algorithm. Additionally, the cut-off date for submissions demonstrates when the classification would occur. $*p < 0.05$, $**p < 0.005$, $***p < 0.0005$, $****p < 0.00005$. Graphs were generated using open-source R and the ggplot2 package under MIT + license (https://cran.r-project.org/web/packages/ggplot2/index.html). Graphs were compiled and the final figure generated using Biorender.com.

(Delta), and P.1 (Gamma)[1]. Our quantitative data analysis supports the exponential emergence of these VOC, with the most emergent being P.1, followed by B.1.617.2, B.1.351, and B.1.1.7, with Pearson's correlation $r$-values of 0.97, 0.96, 0.94, and 0.92, respectively. The quantitative analysis described in this report gives a numerical value to each VOC emergence, predicting the likelihood that the lineage will become prevalent and spread through the population. This value then indicates which VOC genomes are likely to possess evolutionarily selective changes. Previously, using this quantitative analysis, we have demonstrated that the P681H substitution, which had a prevalence of 2% worldwide in December 2020, then emerged to 79% in April 2021[16]. Similarly, using this quantitative analysis in April 2021, we predicted the spread of the Delta variant in Hawaii and worldwide as of June 2021[19].

In this report, we demonstrate the characteristic emergence and selective evolution of VOC using the B.1.1.7 VOC as an example, with a Pearson's correlation of 0.92. The B.1.1.7 VOC, the most prevalent VOC worldwide in April 2021, has spread across the globe after emerging in the United Kingdom in December 2020[21]. As stated, the B.1.1.7 VOC represents the prototypic VOC which has evolved continuously by evading

vaccine sera and becoming more transmissible[22,23]. Similarly, the Delta variant, predicted to be exponentially emerging by this quantitative analysis with an $r$-value of 0.96 as of April 2021 at 1% prevalence, has become the most prevalent worldwide as of June 2021, representing 64% of worldwide sequences.

The virus transmissibility, prevalence, and evasion of both monoclonal antibodies and vaccines are concerning[24]. Thus, the quantitation of VOC leads to identification and quantitation of their respective mutations. The pairwise heat map between variants and mutations (Supplementary Table 2) indicates that SARS-CoV-2 genomes may spontaneously mutate or revert to wildtype, as demonstrated in other statistical analysis for monitoring this pandemic[25].

We were able to establish the algorithm described in this report based on the ten most emerged AAS and deletions. We evaluated nine of the AAS and deletions observed among the variants in this study, as of May 12, 2021, with $r > 0.9$ and >70% prevalence in April 2021. These AAS and deletions included, P681H, ΔV70, ΔH69, N501Y, S982A, D1118H, T716I, A570D, and ΔY144. The data show that the average time for an emerging substitution ($r > 0.9$) to go from >30% monthly prevalence percentage to >50% prevalence is 2.25 months. Extrapolating these findings, the

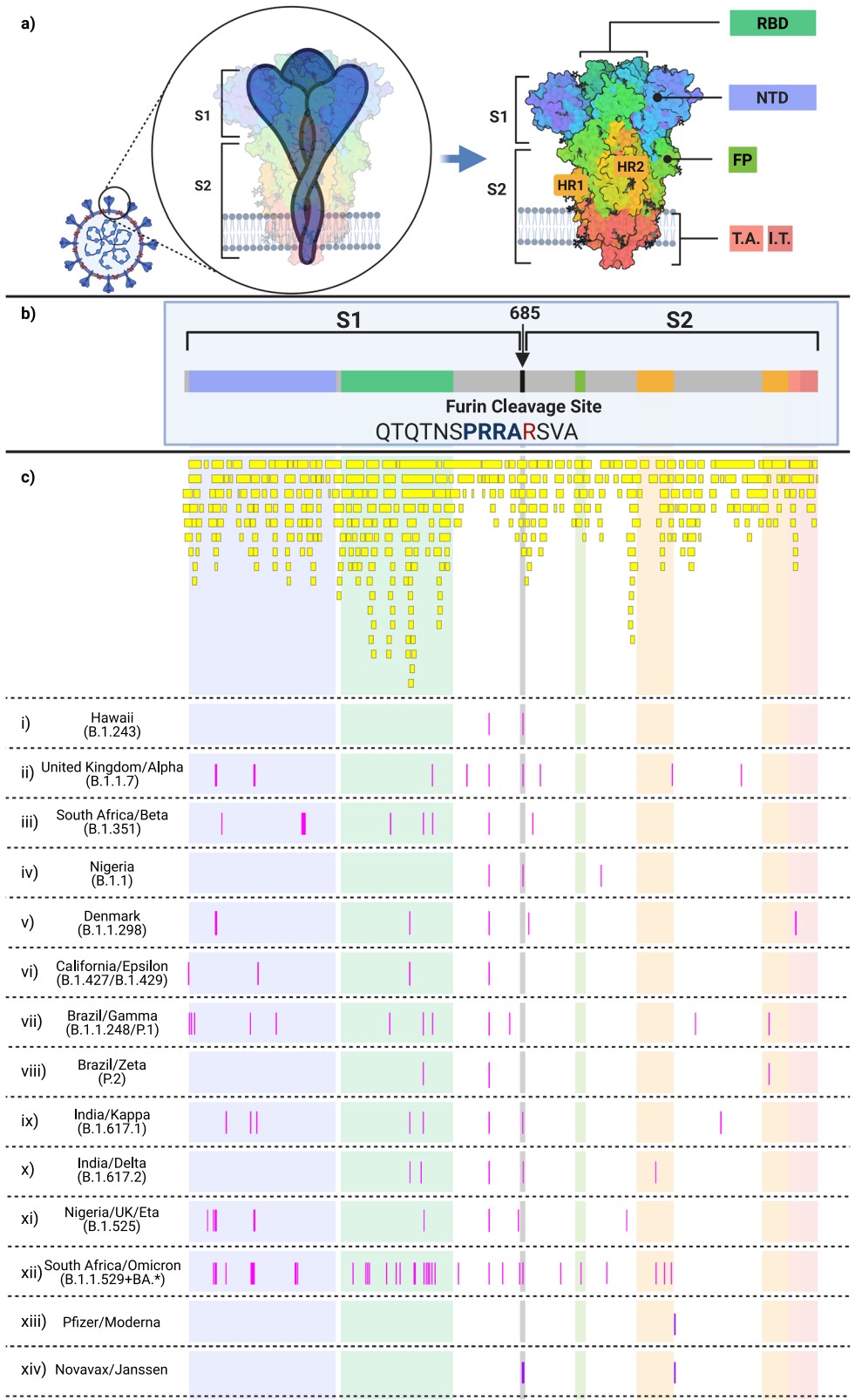

timeframe for Pfizer to identify emerging changes and manufacture them into a new version of the BNT162b2 vaccine would be 60–110 days. This timeframe is roughly equivalent to the algorithm's predictive value[26]. Additionally, the tenth AAS used in the development of the algorithm was D614G that allowed us to discern the previous month prevalence, which is an important parameter, as once an emerging mutation is established in the genome

the $r$ value will decrease considerably. The nine substitutions and deletions also display an average of 4.75 months from >2% monthly prevalence to >50% monthly prevalence. Therefore, $r > 0.9$ and a prevalence of >2% is sufficient to establish a mutation as being of interest, whereas 30% prevalence escalates a mutation to the status of concern. From the evaluation of the ten total spike protein changes, the algorithm concludes that an $r > 0.9$ and a > 30%

**Fig. 4 SARS-CoV-2 spike protein domains and relation to B and T cell epitopes, variant amino acid substitutions, and vaccine amino acid substitutions.** This figure demonstrates the evolution of the SARS-CoV-2 variants by depicting the location of the variants substitutions and deletions in the context of spike domains and epitopes. **a** Cartoon rendering of SARS-CoV-2 and the 1273 amino acid long spike protein overlay onto the color-coded crystallographic structure determined by electron microscopy (PBD ID: 6VXX-PDB). The individual protein domains are color-coded: N-terminal domain (NTD) (light purple) (residues 14-305), receptor-binding domain (RBD) (teal green) (residues 319-541), furin (F) (residues 682-685), fusion protein (FP) (green) (residues 788-806), heptad repeat 1 (HR1) (orange) (residues 912-984), heptad repeat 2 (HR2) (orange) (residues 1163-1213), transmembrane anchor (TM) (light pink) (1213-1237), and intracellular tail domain (IT) (dark pink) (1237-1273). **b** Two-dimensional layout of the spike protein and domains with the addition of the S1/S2 furin cleavage site (RRA/R) (682-685) (black). **c** In silico predicted B and T cell epitope loci revealing 393 in silico B and T cell epitopes mapped here individually as a yellow boxes i–xiii. Amino acid substitutions present in the corresponding variant shown in pink boxes in comparison to the reference sequence NC_045512. (i) B.1.243 Hawaii; (ii) B.1.1.7 United Kingdom; (iii) B.1.351 South Africa; (iv) B.1.1 Nigeria; (v) B.1.1.298 Denmark; (vi) B.1.427 and B.1.429 California; (vii) P.1 Brazil; (viii) P.2 Brazil; (ix) B.1.617.1 India; (x) B.1.617.2 India; (xi) B.1.525 United Kingdom/Nigeria; (xii) B.1.1.529+BA.* South Africa; (xiii) Pfizer and Moderna mRNA sequences with artificially added substitutions K986P and V987P; (xiii) Novavax and Janssen mRNA sequences with artificially added substitutions R682S/Q, R683Q, R685G/Q, K986P, and V987P. Adapted from "An In-depth Look into the Structure of the SARS-CoV-2 Spike Glycoprotein", by BioRender.com (2021). Retrieved from https://app.biorender.com/biorender-templates.

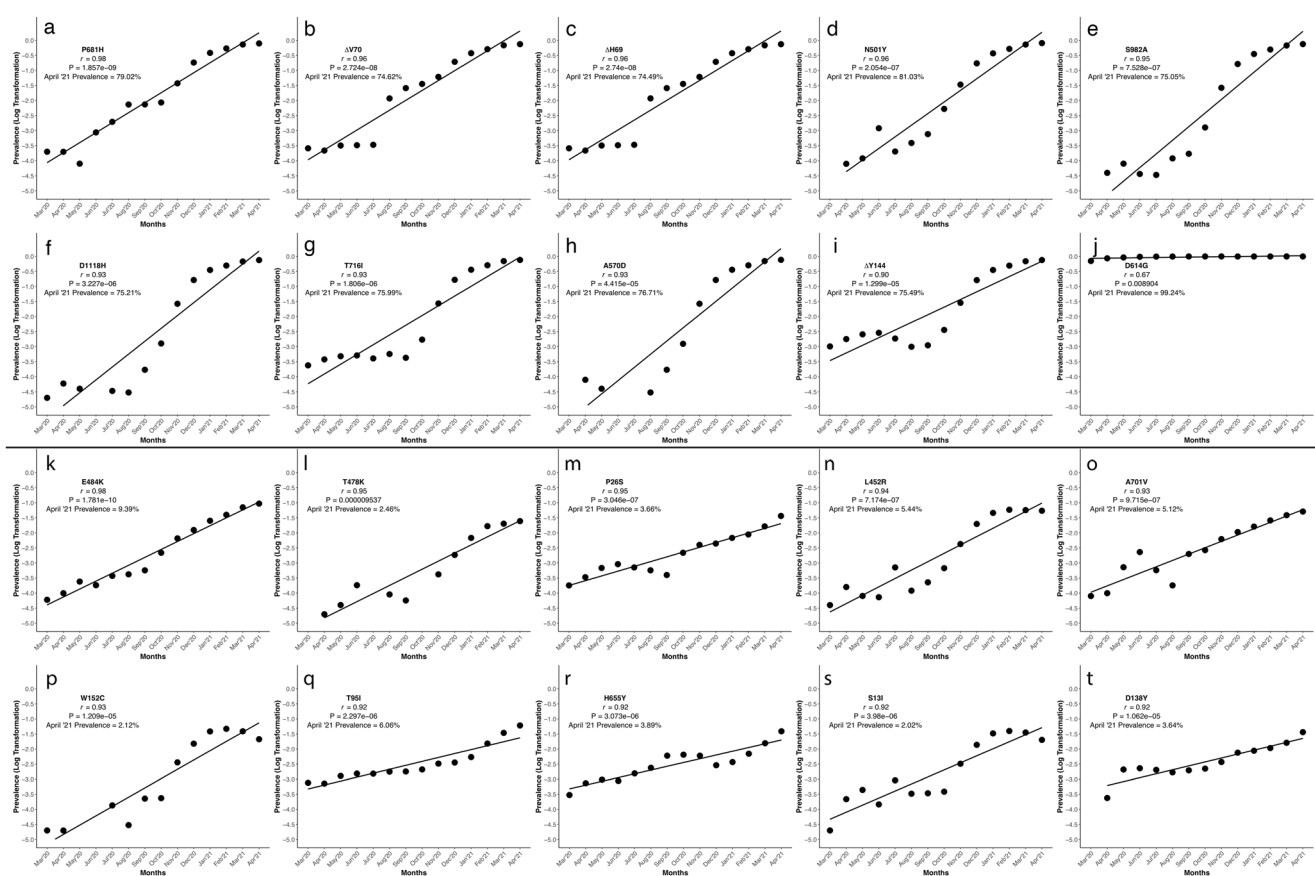

**Fig. 5 Pearson's correlation of logarithmically-transformed prevalence ratios of the most emergent SARS-CoV-2 mutations of concern and interest selected via the algorithm.** This figure shows the graphical representation of the logarithmically-transformed prevalence data used to calculate the Pearson's correlation ($n = 1,483,155$ biologically independent samples) of each of the twenty most emerged (of concern) and emergent (of interest) spike protein substitutions and deletions. The substitutions and deletions of concern here are in order of decreasing $r$ value, and each has a unique alphabet identifier (**a**) P681H, (**b**) ΔV70, (**c**) ΔH69, (**d**) N501Y, (**e**) S982A, (**f**) D1118H, (**g**) T716, (**h**) A570D, (**i**) ΔY144, and (**j**) D614G. The algorithm uses the monthly prevalence data from these ten spike protein substitutions and deletions, and they are the most concerning of all spike changes. The substitutions and deletions of interest here are in order of decreasing $r$ value and each unique substitution or deletion is denoted by a letter of the English alphabet, (**k**) E484K, (**l**) T478K, (**m**) P26S, (**n**) L452R, (**o**) A701V, (**p**) W152C, (**q**) T95I, (**r**) H655Y, (**s**) S13I, and (**t**) D138Y. Graphs were generated using open-source R and the ggplot2 package under MIT + license (https://cran.r-project.org/web/packages/ggplot2/index.html). Graphs were compiled and the final figure created using Biorender.com.

prevalence percentage is an optimal time to classify a substitution as concerning, and consider the substitution for inclusion into vaccine primary structure for 60 day production time.

These mutations of interest and concern can also serve to facilitate and focus research using infectious clones[27], pseudoviruses[28], and deep mutational scanning[11,12] to determine

the functional characteristics and clinical therapeutic consequences. Of great importance in vaccine development is identifying which spike protein changes are most prevalent worldwide across all sequenced genomes. Each substitution or deletion, or combination thereof, could potentially serve as an epitope. Indeed, several substitutions of concern (N501Y) are of interest

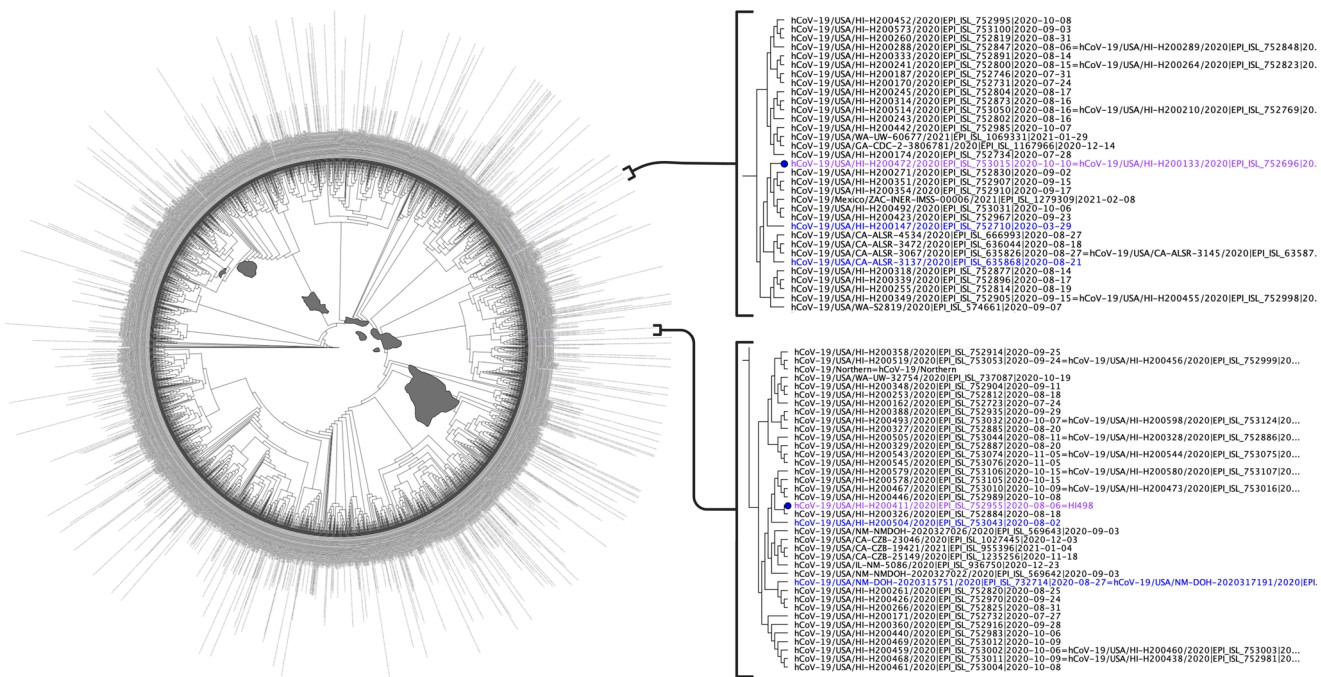

**Fig. 6 Phylogenetic Tree of all B.1.243 Lineage Sequences Worldwide.** This figure displays the phylogenetic tree used to determine the origin of the B.1.243 sequences used in this study. We use 8822 SARS-CoV-2 B.1.243 whole-genome sequences published in the Global Initiative on Sharing Avian Influenza Data (GISAID) and GenBank as of April 12, 2021 to define the origin. From the 8822, 4273 had ambiguous nucleotides between the 5′ and 3′ untranslated regions as determined using multiple alignment using fast Fourier transform (MAFFT). Further, 1588 were duplicate sequences and eight had duplicate identifications as determined by the sRNA Toolbox. Therefore, the final tree was constructed using FastTree in Geneious Prime 2021.1.1 (http://www.geneious.com) from 2953 unique and unambiguous SARS-CoV-2 whole-genome sequences. The HI498 (purple text) origin is defined as New Mexico (blue text) and the HI-708 (purple text) origin is defined as California (blue text). Map prepared with R and usmaps package with a GNU General Public License (GPL), v3 (https://cran.r-project.org/package=usmap). Created with BioRender.com.

(K417N and E484K) shown herein are also involved in monoclonal antibody epitopes as demonstrated in deep mutational scanning of the RBD[11]. Booster vaccines using this algorithm can therefore prepare the vaccinated for any variant they are most likely to encounter by identifying and including the emerging and emerged amino acid changes representing the majority of SARS-CoV-2 worldwide. Deep mutational scanning, in combination with tracking[19,29] and this algorithm, can facilitate public health policies recommending the use or stoppage of a particular mAb when the evasive mutations are spreading, rather than after the therapies begin to fail[10]. In the next versions of this algorithm, the data should be updated every 24 h and evaluated often for recommendations for vaccine design. Such is demonstrated with the improved and more stringent predictive value of the omicron emergence with more frequent partitioning.

To demonstrate the application of our algorithm to the now emerged Delta variant, we draw a comparison between the algorithm and SIG's decision to classify the Delta variant as a VOC. Our algorithm would have classified the Delta as a VOC suitable for adjusting the vaccine on July 01, 2021 ($r = 0.95$, June prevalence = 66% on 07/01/2021 ($n = 1,780,846$ biologically independent samples)), a date only 17 days later than the SIG. Adjusting the algorithm to repartition biweekly rather than monthly (for use in classification) would have given the Delta VOI status on 05/01/2021 and VOC status on the same day as the CDC (June 15, 2021) (02/15/2021 - 06/14/2021; $r = 0.997$, June prevalence = 76% on 06/15/2021 ($n = 1,067,791$ biologically independent samples)). We show that the proposed algorithm is a standardization method on par with the current practice of expert classification and further provides statistical and publicly visible support to those experts. Given the two-month vaccine production time, potential booster vaccines against the Delta variant

could have been available as soon as September 1, 2021. The Delta variant was 95% prevalent worldwide in August 2021 (259,299 of 273,881 reported sequences in the month of August 2021 as of 08/31/2021).

Epitopes are found across nearly the entire spike protein. The in silico compilation of predicted B cell and T cell epitopes demonstrated that 53% (210/393) of all epitopes occur in the NTD and RBD. This is consistent with an in vivo mRNA-LNP vaccine study that found a vast majority of the CD8 + T cell response target epitopes in the N-terminal portion of the Spike protein[30]. The same study found that CD4 + T cell responses target both S1 and S2[30]. Additionally, the majority of variant mutations and neutralizing antibody targets occur in S1[23]. As S1 is shed in the coronavirus model of fusion[31], and S2 is responsible for fusion, the diversity of S1 in SARS-CoV-2 and the epitope targeting concentration in S1, indicate that SARS-CoV-2 vaccines will need to be optimized.

Here, we isolate and characterize SARS-CoV-2 in cell culture and analyze the whole-genome sequences of these isolates and sequences deposited in GISAID to develop an algorithm for next-generation vaccine design. We apply our archetype method for predicting exponentially emerging mutations to these isolates, then evolve the quantitative analysis into an algorithm to evaluate the VOC, VOI, and their mutations, allowing further classification of mutations as concern and interest. Thus, we created a simple and effective vaccine design system that uses the worldwide SARS-CoV-2 sequencing data in a meaningful way. While the sequences are publicly available, there is not yet a defined and logical way to make use of them for next-generation vaccines, monitoring monoclonal antibodies, and adding to public-policy measures for preventing spread of emerging lineages. This algorithm can now serve as a baseline for choosing the primary

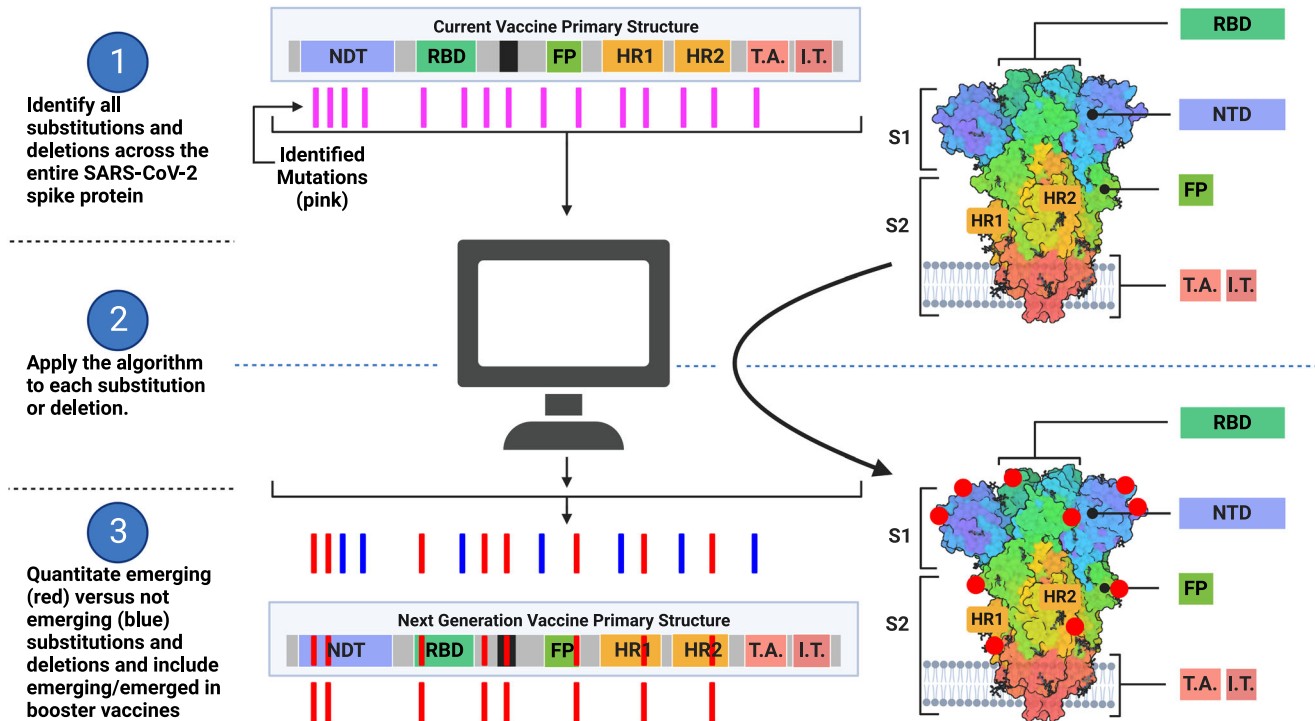

**Fig. 7 Applying the Algorithm to Vaccine Design.** The model displays how the algorithm would lead to the design of next-generation vaccines based on the S gene. Part one of the model identifies all SARS-CoV-2 variants or genomic mutations and spike amino acid substitutions and deletions (represented as pink lines) based on the worldwide sequence databases GISAID and GenBank. The current vaccine design as it translates into proteins is depicted in the top center and top right. Part two of the model will apply the quantitative analysis and algorithm described in this report to each of the protein changes identified in part one. The quantitative analysis determines emergence via logarithmic transformation of prevalence and Pearson's correlation. The algorithm then applies criteria to the quantitative analysis and previous months prevalence for determining which changes are likely to be in the majority of SARS-CoV-2 for incorporation in the next-generation vaccine. Part three of the model determines which substitutions and deletions are exponentially emerging or emerged (red lines) and which are not (blue lines). From part three, the mRNA sequence of vaccines can then incorporate the emerging and emerged mutations so that the folded protein (bottom right) will contain the protein changes (red dots) most prevalent worldwide by the time the next-generation vaccine is manufactured and administered. As a result, these substitutions and deletions will present the most appropriate epitopes of the SARS-CoV-2 spike protein to vaccine recipients. Adapted from "An In-depth Look into the Structure of the SARS-CoV2 Spike Glycoprotein", by BioRender.com (2021). Created with BioRender.com.

structure of vaccines. Additionally, we graphically compare the S gene of the isolates with SARS-CoV-2 VOCs, predict the emergence of SARS-CoV-2 S gene mutations, protein substitutions and deletions, and evaluate them in the context of epitopes. Thus, we create a foundation for future SARS-CoV-2 monitoring and vaccine efforts as we move forward in this pandemic (Fig. 7). These pandemic efforts cannot remain in the context of being responsive and reactive, but need also to be preemptive and predictive. Preemptive and predictive is possible with the approach herein.

These findings have relevance to the future of tracking SARS-CoV-2 and of SARS-CoV-2 vaccine design. The future SARS-CoV-2 vaccines are akin to influenza virus vaccines. That seeming nature is that influenza vaccines change yearly depending on the previous years surveillance data[32]. If the algorithm described herein is instituted, researchers will decipher the evolving SARS-CoV-2 genome in real-time rather than after variants have evolved and emerged.

## Methods

**Patient samples.** Human clinical samples analyzed in this report were part of the University of Hawaiʻi at Mānoa (UHM) approved IRB - H051 study (# 2020-00367) (#NCT04360551) (patient identification [PID] 498 and PID 708) collected as OPS and nasal (NS) swabs at days 5 and 3, following symptom onset, respectively. SARS-CoV-2 positive status was confirmed using quantitative reverse-transcriptase-polymerase chain reaction (qRT-PCR). The samples were stored at −80 °C as part of the UHM IBC approved study (# 20-04-830-05). All SARS-CoV-

2 related research was conducted in the University of Hawaii (UH) Institutional Biosafety Committee (IBC) and EHSO annually certified BSL-3 facility at the John A. Burns School of Medicine.

**Virus isolation.** Virus isolation was conducted using Vero E6 (ATCC CRL-1586) cells and from PID 498 OPS collected in the viral transport medium (VTM)[16]. Briefly, following 1 h infection, cells were monitored for cytopathic effect (CPE) using a Cytosmart Microscope monitoring the same location in the flask at 10X zoom. After observing remarkable CPE at 48 h, supernatant was passaged three times in the Vero E6 cells to create a stock virus. Briefly, at the third passage, the cell monolayer and cell supernatant were freeze-thawed three times at −80 C. The supernatant and cell lysate was centrifuged at 5000 rpm for 15 min at 4 C and the supernatant was aliquoted for plaque assay. USA-WA1/2020, virus was provided by Dr. Mukesh Kumar, Georgia State University, obtained from BEI Resources. PID 498 OPS and USA-WA1/2020 virus isolation was quantitated with plaque assay using a double overlay[33]. The first overlay was layered 1 h-post-infection, the final overlay was layered two days-post-infection (dpi), and the plaques were counted three dpi.

**Growth kinetics.** Following isolation of SARS-CoV-2, isolate USA-HI498 2020, a growth kinetics study was conducted by seeding monolayers of Vero E6 cells in 6 well plates at a cell density of $3 \times 10^5$/well one day prior to the assay. For the assay, cells were infected with SARS-CoV-2 USA-WA1/2020 and HI498 2020 isolates. Briefly, on the day of the assay, DMEM with 10% FBS was removed from wells, cells were washed twice with serum-free DMEM, and inoculated with multiplicity of infection (MOI) 0.1 and 1, diluted in 500 μL of DMEM with 2% FBS and incubated at 37 °C and 5% $CO_2$ for 2 h. Following the 2-h adsorption, the supernatant containing virus was removed, and monolayers were washed twice with DMEM with 2% FBS, and further incubated in 2 mL DMEM with 2% FBS at 37 °C and 5% $CO_2$ until supernatant collection at 0, 12, 24 and 48 h[34].

**RNA extraction, qRT-PCR, and genomic equivalence**. For determining Genomic Equivalence, RNA extraction was conducted using the QIAamp® Viral RNA Mini Kit (Qiagen, Cat# 52906) following the manufacturer's instructions and as described previously[16]. RNA extraction was conducted on eight, ten-fold serial dilutions ($10^0$ to $10^{-8}$)[33]. The primers and probes (N1 set, N2 set, and RdRp set) used were described previously[35,36]. A TaqMan® multiplexed qRT-PCR method was used for the N1 and N2 primer sets. The QuantaBio qScript® XLT One-Step qRT-PCR Tough Mix (Cat# 95132) was used to conduct the qRT-PCR on an ABI StepOnePlus™ Real-Time PCR system. A SYBR Green qRT-PCR method was used for the RdRp primer set. The QuantaBio qScript® cDNA Synthesis Kit (Cat# 95047) and QuantaBio PerfeCTa® qPCR ToughMix™, Low ROX™ (Cat# 95114) was used to conduct the qRT-PCR on an ABI StepOnePlus™ Real-Time PCR system.

Standard curves for the SARS-CoV-2 isolates by N1, N2, and RdRp genes were produced by plotting cycle threshold (Ct) values against corresponding plaque forming units (PFU) per mL evaluated by plaque assay for the eight ten-fold serial dilutions of SARS-CoV-2 virus isolates[33]. The standard curve produced from the ten fold-serial dilutions of the virus was used to interpolate the results of the growth kinetics experiment. All qRT-PCR and Genomic Equivalence data were analyzed and visualized using GraphPad Prism 9 Version 9.2.0.

**Whole genome sequencing**. For WGS, RNA was extracted from both the VTM and the aliquoted third passage stock virus, as described above, with the QIAamp® Viral RNA Mini Kit (Qiagen, Cat# 52906) following the manufacturer's instructions, as described previously[16]. Briefly, viral RNA (vRNA) was eluted in 60 μL of the elution buffer. vRNA extraction was confirmed using the Takara RNA LA PCR Kit (CAT #RR012A) and previously reported primer sets[16,37]. RNA was reverse transcribed into cDNA using the Takara RNA LA PCR Kit (Cat #RR012A) according to the manufacturer's protocol but with a reverse transcription time of 90 min. WGS was conducted by the ASGPB Core, UHM. Briefly, libraries prepared as per the manufacturer's protocol (Illumina Document #1000000025416 v09) using Illumina DNA Prep kit (Cat #20018704) and Nextera XT indexes were sequenced using the MiSeq Reagent Kit v3 (600 cycle) (Cat #MS-102-3003) and an Illumina MiSeq sequencer.

**Informatics**. WGS reads were compiled using the UHM MANA High-Performance Computing Cluster (HPC). Raw fastq sequence files were evaluated by the FASTQC program[38] for quality control metrics. After confirming acceptable quality of the overall reads, low-quality sequences were filtered and trimmed from each read with Trimmomatic[39] using paired-end adapter sequence NexteraPE-PE (ILLUMINACLIP:NexteraPE-PE:2:30:10) and a sliding 4 base window evaluating for quality with a PHRED score over 30. Trimmed result quality was confirmed with FASTQC. The trimmed-paired-end reads were then mapped to the NC_045512 reference genome using Bowtie2[40] and variants called with samtools mpileup[41] and transformed from VCF to FASTQ using bcftools and vcfutils[42] and finally converted to FASTA using seqtk. For comparison and validation, the fastq file was also inputted into Geneious Prime 2021.1.1 to produce FASTA files with the coronavirus assembly workflow[43]. The resultant consensus sequence derived from each Hawai'i isolate was submitted to GenBank (SARS-CoV-2, isolate USA-HI498 2020 (Stock Virus: MZ664037) (VTM: OK021552) and SARS-CoV-2, isolate USA-HI708 2020 (Stock Virus: MZ664038) (VTM: OK189251). The lineage of each sequence was determined with the Phylogenetic Assignment of Named Global Outbreak (PANGO) Lineage nomenclature[44–46].

**Sequences and Lineage Searches**. All Hawai'i sequences as of July 28, 2021, from both GenBank and GISAID were downloaded and searched for the presence of potential lineages of concern using PANGO lineage[19,44–46]. Briefly, all sequences were downloaded from GenBank and GISAID, uploaded to the Pangolin Lineage Assigner (pangolin.cog-uk.io), and the output was evaluated for prevalence using Microsoft Excel.

**Variant comparison**. A comparison was conducted to evaluate the mutations in the two Hawai'i B.1.243 isolates compared to 12 other VOC, VOI, and variants as of May 12, 2021. The NCBI SARS-CoV-2 resources genomic reference sequence from Wuhan was used to define the S gene (NC_045512)[47]. Each of the sequences underwent pairwise alignment with NC_045512 to define S gene mutations. The Lineage (B.1.243) sequence selections were SARS-CoV-2 HI498 and HI708. Sequences for VOC, VOI, and other variants that garnered attention throughout this pandemic were selected with criteria of earliest complete collection dates with unambiguous S gene sequences. Lineages used were: B.1.1.7 (United Kingdom VOC, EPI_ISL_601443)[21,48], B.1.1 (Nigeria variant, EPI_ISL_729975)[49,50], B.1.351 (South Africa VOC, EPI_ISL_712081)[51], B.1.1.298 (Denmark variant, EPI_ISL_616802)[52], B.1.427 (California VOI, EPI_ISL_1531901)[53], B.1.429 (California VOI, EPI_ISL_942929)[54], P.1 (Brazil/Japan VOC, EPI_ISL_792680)[55,56], P.2 (Brazil VOI, EPI_ISL_918536)[57], B.1.617.1 (India VOI, EPI_ISL_1372093)[58], B.1.617.2 (India VOC, EPI_ISL_1663516)[59], and B.1.525 (United Kingdom/Nigeria VOI, EPI_ISL_1739895)[60]. Separetely, the lineages of the omicron variant (B.1.1.529+BA.*, B.1.1.529, BA.1, BA.1.1, BA.2, and BA.3)[1] were evaluated for worldwide emergence as classified by VOC or VOI.

**Variants and amino acid substitution/deletions in comparison to epitope mapping of the spike protein**. From the aforementioned variant comparison section, the selected S gene sequences underwent pairwise alignment with NC_045512 in SnapGene, and SNP were identified. The SNP were inputted into the SnapGene sequence feature and Nextclade[61] to determine amino acid substitutions (AAS). Non-synonymous substitutions were confirmed in GISAID using the metadata for each accession number.

Separately, to determine if any amino acid in the entire S protein can be part of a potential epitope, the following search parameter was used in PubMed to locate in silico studies predicting vaccine epitopes to SARS-CoV-2: "((B-cell) OR (B cell)) AND ((T-cell) OR (T cell)) AND (peptide) AND (vaccine epitope) AND ((SARS-CoV-2) OR (COVID-19))." From this search on January 28, 2021, the three most recent articles[62–64] and the three best matching articles[65–67] were selected for further analysis by mapping to the Spike protein. All predicted epitopes able to be searched and defined with SnapGene's "Find Protein Sequence" feature were included. Article overlaps in the systematic review were only included once.

**Statistics and reproducibility**. The PANGO Server was used to identify and confirm the lineage of each of the aforementioned thirteen strains described in the Variant Comparison section[45,46]. The lineages and their collective AAS were identified and individually searched within EpiCoV™ in GISAID for worldwide prevalence from March 2020–April 2021. Each lineage ($n = 1,479,378$) was filtered separately, as were AAS ($n = 1,483,155$). Parameters for selection were sequences that included a full month, day, and year of collection. Each month's prevalence for each lineage and AAS was logarithmically transformed to serve as the $y$ variable in Pearson's correlation ($n = 14$). The $y$ variable was evaluated against the $x$ variable, month as an interval value ($n = 14$), to determine an exponential increase in worldwide emergence[16]. Pearson's was calculated using RStudio version 1.3.1093 (R version 4.0.3) and plotted with the ggplot2 package. The Pearson's correlations for AAS and lineages were then compared in a corresponding pairwise heat map to evaluate if AAS emergence occurs independently of, or in tandem to, lineage emergence.

**Algorithm**. The algorithm herein described was developed from the quantitation of the ten most emerged amino acid substitutions and deletions. The algorithm is as follows:

For AAS, deletions, and lineages, determine Pearson's coefficient by logarithmically transforming monthly prevalence.

if Pearson's $r \geq 0.9$:

 if Previous Month's Prevalence > 0.3:

 Emerging ("of Concern") (Include in Next Generation Vaccine Design)

 else if $0.02 \leq$ Previous Month's Prevalence $\leq 0.3$:

 Emerging ("of Interest") (Evaluate in Deep Mutational Scanning)

 else:

 Not Emerging

else:

 if Previous Month's Prevalence $\geq 0.9$:

 Emerged ("of Concern") (Include in Next Generation Vaccine Design)

 else if Previous Month's Prevalence > 0.5:

 Emerged ("of Interest") (Evaluate in Deep Mutational Scanning)

 else:

 Not Emerged/Emerging

This algorithm classifies AAS and deletions into two categories based on Pearson's $r$-value, $r \geq 0.9$ and $r < 0.9$. For AAS and deletions to be called out as concerning, the r-value should be $\geq 0.9$ and the previous month's worldwide prevalence of these AAS and deletions should be >30%. Further, these concerning AAS and deletions can be considered for inclusion in the next-generation vaccine design. If the $r \geq 0.9$ and the previous month's prevalence is between 2 and 30%, then the AAS or deletion is classified as interesting, and needs to be evaluated in a research setting. The same algorithm can also be applied for standardizing classifications of SARS-CoV-2 lineages as of interest or of concern.

If the $r < 0.9$, then the focus is on previous month's prevalence of AAS and deletions, as after a mutation is established, there is no longer need to evaluate emergence. If the previous month's prevalence is $\geq 90\%$, then the mutation is established in the SARS-CoV-2 genome and should be considered as concerning and be part of the next-generation vaccine. If the previous month's prevalence is >50%, then the mutation represents the majority, and needs to be considered as interesting and evaluated in a research setting. Again, the same algorithm can also be applied for standardizing classifications of SARS-CoV-2 lineages as of interest or concerning.

**B.1.243 Phylogeny and origin tracking**. Origin tracking was accomplished using an established method[19]. Briefly, all B.1.243 sequences were downloaded from GenBank and GISAID, aligned using using fast Fourier transform (MAFFT) (https://mafft.cbrc.jp/alignment/server/add_fragments.html?frommanualnov6), trimmed of 3' and 5' untranslated regions, removed of ambiguous and duplicate sequences using sRNA toolbox (https://arn.ugr.es/srnatoolbox/helper/removedup/), and an approximately maximum-likelihood phylogenetic tree rooted with Ancestral Lineage A (EPI_ISL_406801) and B (MN908947) was generated using Geneious Prime (http://

www.geneious.com). The sequences from Hawaii were obtained through both GenBank and GISAID, along with SARS-CoV-2, isolate USA-HI498 2020 and SARS-CoV-2, isolate USA-HI708 2020 (from this study), to accomplish the origin determination.

**Reporting summary**. Further information on research design is available in the Nature Research Reporting Summary linked to this article.

## Data availability

The source data used to generate Figs. 1e–m, 2, 3, 5, Supplementary Fig. 1 and Supplementary Table 2 is available in Supplementary Data 1. All GenBank deposition IDs as part of this study are: SARS-CoV-2, isolate USA-HI498 2020 (Stock Virus: MZ664037) (VTM: OK021552) and SARS-CoV-2, isolate USA-HI708 2020 (Stock Virus: MZ664038) (VTM: OK189251). All other data are available from the corresponding author on reasonable request.

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

## Acknowledgements

This research was supported by a grant (P30GM114737) from the Pacific Center for Emerging Infectious Diseases Research, COBRE, a grant (P20GM103466-20S1) from the INBRE, National Institute of General Medical Sciences, and a grant (U54MD007601) from Ola Hawaii, National Institute on Minority Health and Health Disparities, NIH. The H051 clinical trial is registered at ClinicalTrials.gov (#NCT04360551). Computation was supported by NSF grant #1920304 on the University of MANA High Performance Computing Cluster. We thank Dr. Jennifer Saito at the Advanced Studies in Genomics, Proteomics and Bioinformatics (ASGPB) Core, UHM for assistance with WGS.

## Author contributions

Clinical studies: C.M.S.; Conceptualization: D.P.M. and V.R.N.; Data curation: D.P.M., L.L.C., A.C.T., E.N.; Formal analysis: D.P.M., L.L.C., S.B.C., and V.R.N.; Funding acquisition: V.R.N. Investigation: D.P.M. and V.R.N.; Project administration: V.R.N.; Resources: D.P.M. and V.R.N.; Software: D.P.M. and S.B.C.; Supervision: V.R.N.; Validation: D.P.M. and A.C.T.; Visualization: D.P.M. and L.L.C.; Writing - original draft preparation: D.P.M. and V.R.N; Writing - review and editing: D.P.M., L.L.C., S.B.C., A.C.T., E.N., C.M.S., and V.R.N.;

## Competing interests

The authors declare no competing interests.
