## [Peer Review File · Communications Biology]

Reviewers' comments:

Reviewer #1 (Remarks to the Author):

The present manuscript authored by Maison et al. developed a timely algorithm that selects significant and emerging mutations of COVID-19, which can inform next-generation vaccine design. The authors described the bioinformatics pipeline for the detection of SARS-CoV-2 lineage for two patients in Hawaii, and also proposed an algorithm that used mutation data from NCBI and GISAID to propose emerging mutations. However, the implementation of the algorithm needs to be addressed to support active vaccine design research.

Comments:

1. A limitation (or could be extended into a strength) of the algorithm is that the current output (AAS and deletion) do not distinguish mutations that potentially affect antibody recognition, or viral fitness in the host (infectious), etc.? Some mutations could guide better vaccine design, some mutations could inform better public health policies, and some could help design better diagnostic methods. However, in this manuscript, these mutations are described to only support vaccine design.

2. Although the authors mapped the mutations to potential T/B cell epitopes, it appears to me that this type of epitope analysis was done separately from the algorithm. If the epitope analysis was also included in the computation pipeline, please specify. If not, it would be of great interest to also perform epitope analysis for mutations annotated as "Emerging" from the algorithm output.

3. Unlikely other diseases, the current COVID-19 pandemic is an actively changing situation. Reporting significant mutations in a paper-by-paper fashion could potentially limit the prompt response to emerging mutations. Therefore, in order to further utilize this algorithm for vaccine design, it would be of great interest to either set up an online interface (e.g., GitHub) or a command-line tool (using Docker for example). It would also be of interest to discuss how the computational framework could adapt to periodically update from external databases and reports significant mutations.

4. The algorithm can help identify mutations that are already reported (e.g., GISAID), and determine if certain mutations are becoming prevalent. However, it would be of interest to compare/include the SARS-CoV-2 mutations reported by deep mutation scanning (Starr et al., Cell, 2020 and Starr et al., Science, 2021), or the recent global study (Hastie et al., Science, 2021).

Reviewer #2 (Remarks to the Author):

The authors describe development of an algorithm that can be used to predict dominance of a particular SARS-CoV-2 strain within a region, and the appropriateness of a particular vaccine to control the virus. Though others have attempted similar predictive models, the approach presented in this MS is novel. The body of work is substantial, and the end-product could be a useful tool for epidemiologists worldwide.

As written, though the contents were understandable, the MS is not an easy read. This will diminish the value of what could be a 'blockbuster' MS. The following comments are not meant to be a criticism of the science; rather, the authors should consider editing or modifying the text as noted below:

Grammar is faulty in many parts of the MS. That makes many sentences difficult to comprehend in a few parts of the MS. Also, a few sentences are incomplete. This reviewer finds merit in the work but found the manuscript very difficult to read and understand. Some suggested edits/clarifications are listed below:

*This reviewer could not easily interpret the title. I had to read the MS entirely to understand what the

project was all about. Wording in L102-103 could be used to revise the title.

ABSTRACT

L43: Did the virus 'emerge' worldwide or did it 'spread' worldwide?

L44- 47: Totally unclear. Analyzed SARS-CoV-2 genomes of the clinical samples that had been acquired? Not clear what the authors mean by "...used the most worldwide.."; used them for what? The title of the MS states "Hawai'i", adding further confusion to the narrative. Readers will not understand what this means "...monitoring ...vaccine platform". Monitoring what?

L48: Please complete the sentence: "...exponential emergence'...(of what)?"

L48 – 50. It is not clear whether the authors are referring to nt or amino acid 'mutations'. In line 48, they mention genome evaluation (presumably referring to a nt identity comparison) whereas in L49, the discussion centers on "spike protein changes".

L50: Suggest stating "Pearson's coefficient" r-value upfront, so the reader immediately knows what the discussion is referring to.

L54: "so must" is confusing. The latter part of the sentence is the problem; was the authors' intent to say that monitoring and vaccine design must keep up with virus evolution?

INTRODUCTION

L62-64: The sentence is phrased very awkwardly and it is not clear what the authors are trying to state. "...the SARS-CoV-2 evolution,...and constant". What do the authors mean by 'adaptations'? Adaptations to human cells? What do they mean "apparent and constant"? The words "To give nomenclature" is grammatically incorrect.

L71-72: I suggest the authors identify the 3 vaccines. Later on in the MS, they refer to 4 vaccines. This will confuse the reader.

L76: The protein used in which vaccine design? All of them?

L91: Mention of lineage discrepancy comes out of nowhere! What lineage discrepancy? Identify it. How will that knowledge help "...a greater understanding...".

L92-L93: The statements are bewildering. It is not at all clear what the authors mean.

L97: Archetype of the analysis or the sequence?

METHODS

Virus isolation: This section lacks significant explanation/detail. No mention is made of BSL3 facilities and safety precautions. No mention of source of USA-WA1/2020.

L117: Should this be ref. 14?

L117-119: Monitor the same location in the flask? What if CPE showed up elsewhere? What magnification? What is "significant" CPE? How do the authors know the CPE were caused by SARS-CoV-2 and not another co-infecting pathogen?

L120: Why blind passage the virus 3X? Do the authors assume the virus won't change during passage? How long were the passages?

L121: The reference cited is for dengue virus. If the authors used the same procedure, they at minimum should mention when (dpi) the final overlay was layered.

L129: Awkwardly written and leads to confusion: "...1 virus isolates..".

L130: "Infectious supernatant" is an improper descriptor. Supernatant containing virus?

L133: Reference 14 or 16?

L139: Change "are" to "were".

L142: "Perform" is better word choice in place of "conduct".

L148: Why set a curve with PFU when for each PFU, there will be an excess of virus RNA from non-infectious virions or incompletely packaged vRNA. What is the ratio of infectious to non-infectious particle for the virus preparation?

L155: Write as "...RNA was extracted from third..." The authors certainly did not use the third passage to extract RNA; that does not make sense.

L158: RNA extraction was confirmed? What exactly was measured? Cellular RNA? Please explain. Do the authors mean they confirmed virus RNA was present? The kits the authors use will extract both DNA and RNA. For clarity, the authors should consider writing vRNA if the context is in reference to

SARS-CoV-2 RNA.

L160: "Extension time"? Do the authors mean "incubation period"?

L169: "Lack of technical errors"? That is jargon. What exactly do the authors refer to?

L177: Consensus sequences, since they are not identical, or were the consensus sequences of both isolates identical?

L188: This sentence is unclear. Do the authors refer to their isolates, or to a reference B.1.243 genome (what is the GISAID or GenBank reference sequence?).

L212-213: Suggest specifying which GISAID program was used.

L217: Pearson's coefficient was....

RESULTS

276-277: CPE are visualized at relatively low magnification. Why?

278: How does plaque assay "confirm" only SARS-CoV-2 was isolated?

303-305: Which 13? Then 12....

314: Were other databases queried (ie, not just PubMed)?

DISCUSSION

342-343: In the context of vaccine design? Only? I suggest rephrasing this sentence, which will confuse many if not most readers.

361: "...introduced to..", not "introduced in".

389-390: The sentence is not clear. For example, decrease of treatment referring to less people needing to be treated or less people being treated for COVID?

L395: How can a variant "evolve" without "mutations"? This sentence should be re-written for clarity.

L464: Do the authors mean "reactively" than "responsively"? some readers will ask why a 'responsible' action is bad.

L472-474: The point the authors try to make is not clear. Also, GISAID is headquartered in Germany but to this reviewer's knowledge, is financed by the US CDC.

CONCLUSIONS

L477: Isolate two strains in cell cultures, or do the authors mean they evaluated Hawaiian strains of SARS-CoV-2?

L482: 'The" Hawaii isolate? That means only one.

Reviewer #3 (Remarks to the Author):

Authors claim to develop an algorithm for the measuring VOC for rationally designed vaccines based on SARS-CoV-2 isolates.

While paper looks extensive in its discussion, the described algorithm on pages 10-11, lines 230-247, is based on the Pearson's r-value and previous month's prevalence, and is simple, but not convincing. Is this algorithm the authors invention? I expected to see an existing method/algorithm or a review of same or what is currently used by the CDC/WHO in classifying SARS-CoV-2 variants as a concern or interest, for instance; and what similarities or differences exist between the two methods, to measure the article's novelty. Moreover, the implementation of the algorithm is not clear, creating a disconnect between the methods and results.

Authors should make clear their contributions to knowledge and offer useful insights to the implementation phase of the article, as the paper heavily relies on existing tools and the contribution to knowledge appear slim.

REVIEWER 1:

Comment 1: A limitation (or could be extended into a strength) of the algorithm is that the current output (AAS and deletion) do not distinguish mutations that potentially affect antibody recognition, or viral fitness in the host (infectious), etc.? Some mutations could guide better vaccine design, some mutations could inform better public health policies, and some could help design better diagnostic methods. However, in this manuscript, these mutations are described to only support vaccine design.

Response 1: We thank the reviewer for this comment. We would like to add that the current output also evaluates whole lineages. Further adding public policy and antibody recognition is an excellent suggestion and we have added the following verbiage to the revised manuscript. In terms of designing better diagnostic methods, New England Biolabs uses their API to actively monitor mutational changes to the Artic primer set (<https://primer-monitor.neb.com/>). However, NEB could indeed use this algorithm to preemptively alter their commercial products.

Following underlined sentences are edited in the revised manuscript.

Introduction:

L75: "These epitope alterations are also shown to diminish monoclonal antibody effectiveness.⁸⁻¹⁰"

L96: "To answer the dilemma of redesigning next-generation SARS-CoV-2 vaccines and predicting monoclonal antibody effectiveness in populations, we present and further validate our archetype quantitative analysis¹⁷ for determining the emergence of individual substitutions and deletions, and variants, alike, as an algorithm."

Discussion:

L343: "In this report, we lay the foundation for an adaptive and rational algorithm for monitoring SARS-CoV-2 evolution and quantitating variants in the context of the vaccine design and monoclonal antibody therapeutics. These contexts can further facilitate public health policies preemptively."

Comment 2: Although the authors mapped the mutations to potential T/B cell epitopes, it appears to me that this type of epitope analysis was done separately from the algorithm. If the epitope analysis was also included in the computation pipeline, please specify. If not, it would be of great interest to also perform epitope analysis for mutations annotated as "Emerging" from the algorithm output.

Response 2: We thank the reviewer for these comments. We originally intended the T/B cell epitope mapping to demonstrate that redesigning the entire S protein is necessary in vaccine design because epitopes can be predicted across the entire structure. These comments definitely strengthen the inclusion of the epitope mapping. Please see Figure 3 for the connection between substitutions and deletions evaluated in the algorithm and the epitope mapping.

We have inserted the underlined sentence in the revised manuscript:

“Separately, to determine if any amino acid in the entire S protein can be part of a potential epitope, the following search parameter was used in PubMed to locate *in silico* studies predicting vaccine epitopes to SARS-CoV-2: “((B-cell) OR (B cell)) AND ((T-cell) OR (T cell)) AND (peptide) AND (vaccine epitope) AND ((SARS-CoV-2) OR (COVID-19)).”

Comment 3: Unlikely other diseases, the current COVID-19 pandemic is an actively changing situation. Reporting significant mutations in a paper-by-paper fashion could potentially limit the prompt response to emerging mutations. Therefore, in order to further utilize this algorithm for vaccine design, it would be of great interest to either set up an online interface (e.g., GitHub) or a command-line tool (using Docker for example). It would also be of interest to discuss how the computational framework could adapt to periodically update from external databases and reports significant mutations.

Response 3: We thank the reviewer for this comment. We agree that a paper-by-paper basis is impractical, and setting up an online interface or a command-line tool would be the direction we would like to see this in the future. Currently, our team does not have the expertise or funding to pursue this option. We would hope to establish an app that interfaces with NCBI, GISAID, and potentially the China National Center for Bioinformation. The app would then catalog the lineage and mutations of each accession number and perform the statistics and algorithm. Anyone could then use this app such as the vaccine manufacturers, policy-makers and CDC to redesign or implement polyvalent vaccines, logically decide on quarantine protocols, and statistically validate classification of variants.

We have added the following sentences to the Algorithm Discussion Section:

“Next generation sequencing is a timely process and requires continuous re-evaluation of the data from past collection dates. In the next versions of this algorithm, the data should be updated every 24 hours and evaluated often for recommendations for vaccine design.”

Comment 4: The algorithm can help identify mutations that are already reported (e.g., GISAID), and determine if certain mutations are becoming prevalent. However, it would be of interest to compare/include the SARS-CoV-2 mutations reported by deep mutation scanning (Starr et al., Cell, 2020 and Starr et al., Science, 2021), or the recent global study (Hastie et al., Science, 2021).

Response 4:

We thank the reviewer for these most interesting comments and insights. We have added the following to the algorithm and discussion to incorporate these ideas.

L238 & L235: “Emerging (of Interest) (Evaluate in Deep Mutational Scanning)”

L420: “These mutations of interest and concern can also serve to facilitate and focus research using infectious clones⁶², and pseudoviruses⁶³, and deep mutational scanning^{9,10} to determine the functional characteristics and clinical therapeutic consequences.”

L427: “Indeed, several substitutions of concern (N501Y) are of interest (K417N and E484K) shown herein are also involved in monoclonal antibody epitopes as demonstrated in deep mutational scanning of the RBD.⁹⁹”

L430: “Deep mutational scanning, in combination with tracking and this algorithm, can facilitate public health policies recommending the use or stoppage of a particular mAb when the evasive mutations are spreading, rather than after the therapies begin to fail.”⁸”

REVIEWER 2:

Comment 1: *This reviewer could not easily interpret the title. I had to read the MS entirely to understand what the project was all about. Wording in L102-103 could be used to revise the title.

Response 1: We thank the reviewer for this comment. Based on reviewer 2 comments 1 and 3 we are modifying the title to reflect the central message of this manuscript. The revised title is “Dynamic SARS-CoV-2 Emergence Algorithm for Rationally-Designed Logical Next-Generation Vaccines”.

Comment 2: L43: Did the virus ‘emerge’ worldwide or did it ‘spread’ worldwide?

Response 2: We thank the review for this comment and have made appropriate changes to the revised manuscript.

Comment 3: L44- 47: Totally unclear. Analyzed SARS-CoV-2 genomes of the clinical samples that had been acquired? Not clear what the authors mean by “...used the most worldwide..”; used them for what? The title of the MS states “Hawai’i”, adding further confusion to the narrative. Readers will not understand what this means “..monitoring ...vaccine platform”. Monitoring what?

Response 3: We thank the reviewer for these comments. The sentence has been changed in the revised manuscript to more accurately reflect the approach.

“We acquired SARS-CoV-2 positive clinical samples and compared the worldwide emerged spike gene mutations from Variants of Concern/Interest, and developed an algorithm for monitoring the evolution of SARS-CoV-2 in the context of vaccines and monoclonal antibodies.”

Comment 4: L48: Please complete the sentence: “...exponential emergence’...(of what)?

Response 4: We thank the reviewer for this comment. The sentence has been edited to reflect the clarity needed.

“The algorithm partitions logarithmic-transformed prevalence data monthly and Pearson’s correlation determines exponential emergence of substitutions and lineages.”

Comment 5: L48 – 50. It is not clear whether the authors are referring to nt or amino acid ‘mutations’. In line 48, they mention genome evaluation (presumably referring to a nt identity comparison) whereas in L49, the discussion centers on “spike protein changes”.

Response 5: We thank the reviewer for catching this error. Please note we have accepted the suggestion and edited the sentence. The new sentence is:

“The SARS-CoV-2 genome evaluation indicated 49 substitutions.”

Comment 6: L50: Suggest stating “Pearson’s coefficient” r-value upfront, so the reader immediately knows what the discussion is referring to.

Response 6: We thank the reviewer for this suggestion, which has been incorporated into the manuscript. The edited sentence is as follows with edits underlined:

“Nine of the ten most worldwide prevalent (>70%) spike protein changes have Pearson’s coefficient *r*-values >0.9.”

Comment 7: L54: “so must” is confusing. The latter part of the sentence is the problem; was the authors’ intent to say that monitoring and vaccine design must keep up with virus evolution?

Response 7: We thank the reviewer for this suggestion. Yes, monitoring the substitutions and vaccine design must keep up with virus evolution. Therefore in consideration of this comment and comment from Reviewer 1, we have incorporated the following sentence in the revised manuscript.

“Monitoring, next-generation vaccine design, and mAb clinical efficacy must keep up with SARS-CoV-2 evolution, as the virus is predicted to remain endemic.”

Comment 8: L62-64: The sentence is phrased very awkwardly and it is not clear what the authors are trying to state. “...the SARS-CoV-2 evolution,...and constant”. What do the authors mean by ‘adaptations’? Adaptations to human cells? What do they mean “apparent and constant”? The words “To give nomenclature” is grammatically incorrect.

Response 8: We thank the reviewer for the questions and comments. However, since our meaning was unclear, we have edited the sentences and the following sentences are inserted in the revised manuscript.

“From the establishment of the now universal D614G substitution³ to the emergence of the VOC and VOI with dozens of different mutations across their respective genomes,¹ SARS-CoV-2 evolution has been evident throughout the pandemic. To define these evolutionary events, the Centers for Disease Control and Prevention (CDC) has classified certain lineages as VOC and VOI to denote highly adapted and immunologically evasive strains of SARS-CoV-2 based on expert evaluation of available data by the SARS-CoV-2 Interagency Group.^{1”}

Comment 9: L71-72: I suggest the authors identify the 3 vaccines. Later on in the MS, they refer to 4 vaccines. This will confuse the reader.

Response 9: We thank the reviewer for this comment. We have identified the 3 vaccines. However, we were unable to find any reference to four vaccines. We do refer to four variants of concern several times, as at the time of writing the algorithm, the CDC had identified four variants of concern. The L71-72 sentence has been amended as follows:

“In the United States, three of these vaccines (Pfizer and BioNtech BNT162b2, Moderna mRNA-1273, and Janssen Ad26.COVS) are authorized and recommended by the U.S. Food and Drug Administration (FDA).^{5,6”}

Comment 11: L76: The protein used in which vaccine design? All of them?

Response 11: We thank the reviewer for this comment. To further clarify, we have edited the sentence as follows:

“Several of these mutations are found in the spike protein used in vaccine design, and therefore allows the virus to evade antibodies targeted to the original strain of that vaccine.”

Comment 12: L91: Mention of lineage discrepancy comes out of nowhere! What lineage discrepancy? Identify it. How will that knowledge help "...a greater understanding...".

Response 12: We thank the reviewer for this comment. We have relocated and refined the paragraph, and removed the words "lineage discrepancy." Additionally, we have clarified the meaning we wished to portray, "the diverse ethnic population of Hawaii being affected disproportionately by COVID-19. The revised paragraph in the revised manuscript is as follows: "Hawai'i has been disproportionately affected by COVID-19 in terms of race, wherein 20% of the cases occur in 4% of the population of Pacific Islanders.^{14,15} Understanding the SARS-CoV-2 as it relates to emerging lineages in Hawai'i, a isolated island community with diverse ethnic groups, will allow for a greater understanding of the pandemic's nature worldwide."

Comment 13: L92-L93: The statements are bewildering. It is not at all clear what the authors mean.

Response 13: We thank the reviewer for this comment. We have relocated and refined the paragraph as described in Response 12.

Comment 14: L97: Archetype of the analysis or the sequence?

Response 14: We thank the reviewer for this comment. To further clarify, we have edited the sentence as follows:

"To answer the dilemma of redesigning next-generation SARS-CoV-2 vaccines and predicting monoclonal antibody effectiveness in populations, we present and further validate our previously described quantitative analysis¹⁷ for determining the emergence of individual substitutions and deletions and variants, alike, as an algorithm."

Comment 15: Virus isolation: This section lacks significant explanation/detail. No mention is made of BSL3 facilities and safety precautions. No mention of source of USA-WA1/2020.

Response 15: We thank the reviewer for this comment. To further clarify, we have edited the sentences as follows:

"All virus research was conducted in the certified BSL-3 facility at the University of Hawaii John A. Burns School of Medicine"

"USA-WA1/2020 was provided by Dr. Mukesh Kumar, Georgia State University, obtained from BEI Resources. PID 498 OPS and USA-WA1/2020 virus isolation was quantitated with plaque assay using a double overlay, performed as described previously"

Comment 16: L117: Should this be ref. 14?

Response 16: We thank the reviewer for catching this error. Yes, this should be ref. 14. This has been updated in the revised manuscript.

Comment 17: L117-119: Monitor the same location in the flask? What if CPE showed up elsewhere? What magnification? What is "significant" CPE? How do the authors know the CPE were caused by SARS-CoV-2 and not another co-infecting pathogen?

Response 17: We thank the reviewer for these important comments. The same location is monitored since we attach (fix) the flask to the microscope inside the incubator. Significant meaning near complete CPE. We do not know if other co-infecting pathogens are present in the

OPS. That said, the patient was clinically diagnosed as having COVID-19, which was further validated by OPS based realtime TaqMan® PCR. Further, whole genome sequencing of the viral RNA extracted from the VTM used for virus isolation, identified the presence of SARS-CoV-2 in the VTM.

Comment 18: L120: Why blind passage the virus 3X? Do the authors assume the virus won't change during passage? How long were the passages?

Response 18: We thank the reviewer for this comment. The virus was passaged to create stock virus for future experiments. To address the potential of nucleotide sequence change during passaging, we also conducted whole genome sequencing on the original VTM and the stock virus. There was no change in the sequences between the VTM and the stock virus. The whole genome sequences for both the original VTM (OK021552) and the stock virus (MZ664037) are deposited in the GenBank. Further, the stock virus, SARS-CoV-2, Isolate USA-HI498 2020, GenBank Accession Number MZ664037, is deposited in BEI Resources (Catalog Number NR-56130). We have edited the following sentences in the revised manuscript:

Methods:

“After observing significant CPE at 48 hours, supernatant was passaged three times in the Vero E6 cells to create a stock virus. USA-WA1/2020, virus was provided by Dr. Mukesh Kumar, Georgia State University, obtained from BEI Resources. PID 498 OPS and USA-WA1/2020 virus isolation was quantitated with plaque assay using a double overlay, performed as described previously.”

“The resultant consensus sequence was defined as Hawai'i Isolates and submitted to GenBank (SARS-CoV-2, Isolate USA-HI498 2020 (Stock Virus: MZ664037) (VTM: OK021552) and SARS-CoV-2, Isolate USA-HI708 2020 (Stock Virus: MZ664038) (VTM: OK189251)).”

“For WGS, RNA extraction was conducted using both the VTM and the third blind passage with the QIAamp® Viral RNA Mini Kit (Qiagen, Cat# 52906) following the manufacturer's instructions, as described previously.”

Results:

“SARS-CoV-2, Isolate USA-HI498 2020 (GenBank Accession Number MZ664037) is deposited in BEI Resources Cat# NR-56130.”

“There was no change in the sequences between the VTM and the stock virus. The whole genome sequences for both the original VTM (OK021552) and the stock virus (MZ664037) are deposited in the GenBank.”

Comment 19: L121: The reference cited is for dengue virus. If the authors used the same procedure, they at minimum should mention when (dpi) the final overlay was layered.

Response 19: We thank the reviewer for this comment. We have added the following sentence to the revised manuscript:

“The first overlay was layered one hour-post-infection, the final overlay was layered two days-post-infection (dpi), and the plaques were counted three dpi.”

Comment 20: L129: Awkwardly written and leads to confusion: “...1 virus isolates..”.

Response 20: We thank the reviewer for this comment. We have edited the sentence.

“Briefly, on the day of the assay, DMEM in 10% FBS was removed from wells with Vero cells, wells were washed twice with serum-free DMEM, and inoculated with multiplicity of infection (MOI) 0.1 and 1, diluted in 500 μ L of DMEM in 2% FBS and incubated at 37°C and 5% CO₂ for two hours.”

Comment 21: L130: “Infectious supernatant” is an improper descriptor. Supernatant containing virus?

Response 21: We thank the reviewer for this comment. We have edited the sentence.

“Following the two-hour adsorption, the supernatant containing virus was removed, and monolayers were washed twice with DMEM in 2% FBS, and further incubated in 2 mL DMEM with 2% FBS at 37°C and 5% CO₂ until supernatant collection at 0, 12, 24, and 48 hours.”

Comment 22: L133: Reference 14 or 16?

Response 22: We thank the reviewer for catching this error. The reference is corrected, Lednicky *et al.*, 2020 (doi:10.4209/aaqr.2020.05.0202).

Comment 23: L139: Change “are” to “were”.

Response 23: We thank the reviewer for this comment. We agree with the reviewer and have made the change.

Comment 24: L142: “Perform” is better word choice in place of “conduct”.

Response 24: We thank the reviewer for this suggestion. We agree with the reviewer and have made the change.

Comment 25: L148: Why set a curve with PFU when for each PFU, there will be an excess of virus RNA from non-infectious virions or incompletely packaged vRNA. What is the ratio of infectious to non-infectious particle for the virus preparation?

Response 25: We thank the reviewer for this comment. Standard curve was generated to quantitate and correlate the Ct values with plaque forming units as described previously by Johnson *et al.*, 2005 (doi:10.1128/JCM.43.10.4977-4983.2005). We do not have access to the technique to determine the ratio of infectious to non-infectious virus particles.

Comment 26: L155: Write as “..RNA was extracted from third...” The authors certainly did not use the third passage to extract RNA; that does not make sense.

Response 26: We thank the reviewer for this comment. As mentioned previously, we extracted RNA from the original VTM and the third passage stock virus for whole genome sequences (WGS). The sequences from both samples were identical.

Comment 27: L158: RNA extraction was confirmed? What exactly was measured? Cellular RNA? Please explain. Do the authors mean they confirmed virus RNA was present? The kits the authors use will extract both DNA and RNA. For clarity, the authors should consider writing vRNA if the context is in reference to SARS-CoV-2 RNA.

Response 27: We thank the reviewer for this comment. We have included the suggestion by adding the following underlined words to the revised manuscript.

“ Briefly, viral RNA (vRNA) was eluted in 60 µL of the elution buffer. vRNA extraction was confirmed using the Takara RNA LA PCR Kit (CAT #RR012A) and previously reported primer sets.”

Comment 28: L160: “Extension time”? Do the authors mean “incubation period”?

Response 28: We thank the reviewer for this comment. We mean “reverse transcription.” The edited sentence is inserted in the revised manuscript.

“RNA was reverse transcribed into cDNA using the Takara RNA LA PCR Kit (Cat #RR012A) according to the manufacturer's protocol but with a reverse transcription time of 90 minutes.”

Comment 29: L169: “Lack of technical errors”? That is jargon. What exactly do the authors refer to?

Response 29: We thank the reviewer for this comment. We have edited the sentence in the revised manuscript.

“Raw fastq sequence files were evaluated by the FASTQC program for quality control metrics. After confirming acceptable quality of the overall reads, low-quality sequences were filtered and trimmed from each read with Trimmomatic using paired-end adapter sequence NexteraPE-PE (ILLUMINACLIP:NexteraPE-PE:2:30:10) and a sliding 4 base window evaluating for quality with a PHRED score over 30.”

Comment 30: L177: Consensus sequences, since they are not identical, or were the consensus sequences of both isolates identical?

Response 30: We thank the reviewer for this comment. The whole-genome sequences of two Hawaii isolates are not identical. We have accordingly edited the sentence in the revised manuscript.

“The resultant consensus sequence derived from each Hawaii isolate was submitted to GenBank (SARS-CoV-2, Isolate USA-HI498 2020 (MZ664037) (VTM: OK021552) and SARS-CoV-2, Isolate USA-HI708 2020 (MZ664038) (VTM: OK189251)).”

Comment 31: L188: This sentence is unclear. Do the authors refer to their isolates, or to a reference B.1.243 genome (what is the GISAID or GenBank reference sequence?).

Response 31: We thank the reviewer for this comment. Our two Hawaii isolates are lineage B.1.243.

“A comparison was conducted to evaluate the mutations in the two Hawaii B.1.243 isolates compared to 12 other VOC, VOI, and variants as of May 12, 2021. ”

Comment 32: L212-213: Suggest specifying which GISAID program was used.

Response 32: We thank the reviewer for this comment. We have updated the manuscript to include the program.

“The lineages and their collective AAS were identified and individually searched within EpiCoV™ in GISAID for worldwide prevalence from March 2020 - April 2021.”

Comment 33: L217: Pearson’s coefficient was....

Response 33: We thank the reviewer for identifying our lack of clarity. Here, we are describing the data used to conduct Pearson’s correlation analysis: x = month as an interval value, y = logarithmic-transformed monthly prevalence. We have rewritten this section to reflect this change.

“Each month’s prevalence for each lineage and AAS was logarithmically transformed to serve as the y variable in Pearson’s correlation. The y variable was evaluated against the x variable, month as an interval value, to determine an exponential increase in worldwide emergence as described previously.”

Comment 34: 276-277: CPE are visualized at relatively low magnification. Why?

Response 34: We thank the reviewer for this question. The real-time microscope that we have only functions at 10X magnification, as per the manufacturer’s instructions.

Comment 35: 278: How does plaque assay “confirm” only SARS-CoV-2 was isolated?

Response 35: We thank the reviewer for this comment. Plaque assays do not confirm only SARS-CoV-2 was isolated. Plaque assays quantitate the virus. Our sequencing confirms the isolation of SARS-CoV-2. We have updated the revised manuscript:

“Plaque assay quantitated SARS-CoV-2, Isolate USA-HI498 at 1.28×10^7 PFU/mL and SARS-CoV-2 USA-WA1/2020 at 3.88×10^7 PFU/mL.”

Comment 36: 303-305: Which 13? Then 12....

Response 36: The first sequence is the reference Wuhan sequence. 13 spike sequence total, 12 are variant sequences. We have updated the revised manuscript.

“The output S gene alignment between the thirteen genomic sequences (12 variants and one reference Wuhan sequence) identified 49 SNPs. Pearson’s correlation on logarithmically-transformed prevalence was calculated for the 12 SARS-CoV-2 variants in this study in order of highest to lowest r value as outlined in Table 2 and Figure 2.”

Comment 37: 314: Were other databases queried (ie, not just PubMed)?

Response 37: We thank the reviewer for this question. We only queried the PubMed database for this analysis.

Comment 38: 342-343: In the context of vaccine design? Only? I suggest rephrasing this sentence, which will confuse many if not most readers.

Response 38: We thank the reviewer for this comment. We have rephrased the suggested sentence. We are focusing on next-generation vaccine design as we see that as a necessary step in combating this pandemic, although we agree that the algorithm is dynamic and can serve many purposes. There is no existing algorithm for facilitating SARS-CoV-2 vaccine design. Currently, the CDC experts classify variants and implement public policy. We hope the

CDC and the scientific community sees the same value and potential in this algorithm that the reviewers recognized. We have, therefore, noted in our discussion (based on reviewer 3 comment 2) that our algorithm serves as statistical support to CDC decisions down to the day.

L342-343: “In this report, we lay the foundation for an adaptive and rational algorithm for monitoring SARS-CoV-2 evolution and quantitating variants in the context of vaccine design.”

Comment 39: 361: “..introduced to..”, not “introduced in”.

Response 39: We thank the reviewer for this comment. We have made the edit.

Comment 40: 389-390: The sentence is not clear. For example, decrease of treatment referring to less people needing to be treated or less people being treated for COVID?

Response 40: We thank the reviewer for this comment. We have edited the sentence in the revised manuscript as follows:

“The virus transmissibility, prevalence, and evasion of both monoclonal antibodies and vaccines are concerning.”

Comment 41: L395: How can a variant “evolve” without “mutations”? This sentence should be re-written for clarity.

Response 41: We thank the reviewer for this comment. We have rephrased the sentence in the revised manuscript as follows for clarity.

“The pairwise heat map between variants and mutations (Table 2) indicates that SARS-CoV-2 genomes may spontaneously mutate or revert to wildtype, as demonstrated in other statistical analysis for monitoring this pandemic.”

Comment 42: L464: Do the authors mean “reactively” than “responsively”? some readers will ask why a ‘responsible’ action is bad.

Response 42: We thank the reviewer for this comment. We are saying that the algorithm allows researchers to take action preemptively, rather than after evasive viral evolution has occurred. As this virus is militant, and millions are dying, we felt military terms were appropriate. We have, however, rephrased for clarity.

“If the algorithm described herein is instituted, researchers will decipher the evolving SARS-CoV-2 genome in real-time rather than after variants have evolved and emerged.”

Comment 43: L472-474: The point the authors try to make is not clear. Also, GISAID is headquartered in Germany but to this reviewer’s knowledge, is financed by the US CDC.

Response 43: We thank the reviewer for this comment. We are saying that the vast majority of sequences uploaded in GISAID are different from the sequences uploaded in GenBank. There is no interface between the two platforms. GISAID has useful filters in terms of searching for specific substitutions or variants. As such, to evaluate all the world data in real-time, all the platforms need to be interfaced. In view of the reviewers comment, we have rephrased the paragraph in the revised manuscript and removed the referral to GISAID headquarters. :

“As the number of worldwide SARS-CoV-2 sequences grows into many millions between GenBank and GISAID, the need for an Application Programming Interface (API) between SARS-CoV-2 sequence databases is needed. Such an API would allow the quantitation of emerging mutations and variants to alarm when necessary. Real-time quantitation would then allow vaccines to evolve preemptively. As things stand, the data is self-diversifying by researchers' choice of submission to one database or the other, rendering GISAID's filters less informative on a worldwide scale. Since our algorithm currently uses GISAID's filters, in the future a centralized database will enhance the implementation of the algorithm.”

Comment 44: L477: Isolate two strains in cell cultures, or do the authors mean they evaluated Hawaiian strains of SARS-CoV-2?

Response 44: We thank the reviewer for this comment. We have rephrased for clarity as follows:

“Here, we isolate and evaluate SARS-CoV-2 in cell cultures from patients infected in Hawaii.”

Comment 45: L482: ‘The’ Hawaii isolate? That means only one.

Response 45: We thank the reviewer for this comment. The S gene of the two Hawaii isolates are identical. We have edited the sentence in the revised manuscript.

“Additionally, we graphically compare the S gene of the Hawaii isolates with SARS-CoV-2 VOCs, predict the emergence of SARS-CoV-2 S gene mutations, protein substitutions and deletions, and evaluate them in the context of epitopes.”

REVIEWER 3:

Comment 1: While paper looks extensive in its discussion, the described algorithm on pages 10-11, lines 230-247, is based on the Pearson's r-value and previous month's prevalence, and is simple, but not convincing. Is this algorithm the authors invention?

Response 1: We thank the reviewer for this comment. We, too, are pleased at how something as simple as Pearson's correlation on logarithmic-transformation prevalence has been so practical. In our recent publication, we predicted the emergence of the Alpha Variant of Concern via predicting the emergence of the P681H substitution (Maison et al., 2021; PMID: 33442699, PMID: 33718878). The current manuscript predicts the emergence of the Delta variant by classifying it as more emerging than the Alpha, Beta, Epsilon, Kappa, Eta, and Zeta Variants. Yes, the algorithm is our invention based upon the statistical analysis from our previous publication (Maison et al., 2021; PMID: 33442699, PMID: 33718878).

Comment 2: I expected to see an existing method/algorithm or a review of same or what is currently used by the CDC/WHO in classifying SARS-CoV-2 variants as a concern or interest, for instance; and what similarities or differences exist between the two methods, to measure the article's novelty. Moreover, the implementation of the algorithm is not clear, creating a disconnect between the methods and results.

Response 2: We thank the reviewer for these comments. We respectfully point out that this algorithm is for facilitating vaccine design, but is dynamic with possibilities in standardizing variant classification. Currently, CDC/WHO does not have a standardized method for assigning these classifications. The variant classification decision is made by the US Department of Health

and Human Services SARS-CoV-2 Interagency Group (SIG) on the following criteria: “The SIG meets regularly to evaluate the risk posed by SARS-CoV-2 variants circulating in the United States and to make recommendations about the classification of variants. This evaluation is undertaken by a group of subject matter experts who assess available data, including variant proportions at the national and regional levels and the potential or known impact of the constellation of mutations on the effectiveness of medical countermeasures, severity of disease, and ability to spread from person to person.”

(<https://www.cdc.gov/coronavirus/2019-ncov/variants/variant-info.html>) The frequency of SIG meetings is not specified.

Our algorithm is the first proposed next-generation vaccine design method, which can additionally be used for standardizing classifications, and monitoring monoclonal antibodies in the context of public health policies.

Table 2 demonstrates the implementation of the algorithm, whereby both previous month prevalence and Pearson’s coefficient r -value are shown side-by-side for both lineages and amino acid substitutions. In terms of implementation within the pandemic itself would require the SIG to recognize and utilize the algorithm. The actual algorithm is written in Python programming language, but would require far more complex programming than is our expertise. We would have to find collaborators and funding to further develop and implement the complex programming aspect of this algorithm.

In view of the reviewers comment, we have added the following to the revised manuscript:

The following paragraph is added in the Discussion section:

“To demonstrate the application of our algorithm to the now emerged Delta variant, we draw a comparison between the algorithm and SIG’s decision to classify the Delta variant as a VOC. Our algorithm would have classified the Delta as a VOC suitable for adjusting the vaccine on July 01, 2021 ($r = 0.95$, June prevalence = 66% on 07/01/2021), a date only 17 days later than the SIG. Adjusting the algorithm to repartition biweekly rather than monthly (for use in classification) would have given the Delta VOC status on the same day as the CDC (June 15, 2021) (06/01/2021 - 06/14/2021; $r = 0.95$, June prevalence = 76% on 06/15/2021). We show that the proposed algorithm is a standardization method on par with the current practice of expert classification and further provides statistical and publicly visible support to those experts. Given the two-month vaccine production time, potential booster vaccines against the Delta variant could have been available as soon as September 1, 2021. The Delta variant was 95% prevalent worldwide in August 2021 (703,742 of 738,866 reported sequences in the month of August 2021).”

The following sentence is added in the Introduction section:

L67: “To define these evolutionary events, the Centers for Disease Control and Prevention (CDC) has classified certain lineages as VOC and VOI to denote highly adapted and

immunologically evasive strains of SARS-CoV-2 based on expert evaluation of available data by the SARS-CoV-2 Interagency Group.¹”

The following sentence is added in the Results section.

“To demonstrate the implementation of the algorithm, in Table 3, we have shown the unique nucleotide mutations and resulting AAS and deletions for each of the twelve SARS-CoV-2 variants, in comparison to the reference sequence.”

Comment 3: Authors should make clear their contributions to knowledge and offer useful insights to the implementation phase of the article, as the paper heavily relies on existing tools and the contribution to knowledge appear slim.

Response 3: We thank the reviewer for this comment. Our contribution to knowledge is creating a simple and effective vaccine design system that uses the worldwide sequencing data in a meaningful way. While the sequences are available, there is not yet a defined and logical way to make use of them for next-generation vaccines, monitoring monoclonal antibodies, and adding to public-policy measures preventing spread of emerging lineages. Thus, we present a practical algorithm that we invented from our previous work (Maison et al., 2021; PMID: 33442699, PMID: 33718878) that gives standardization to logical next-generation SARS-CoV-2 vaccine design in the future. As the other reviewers have also noted, there are several dynamic applications of the algorithm, from precision public health genomic evaluation of monoclonal antibodies to supporting CDC variant classification. To make our contribution to knowledge clear, we have added the following sentence in the conclusion section of the revised manuscript.

“Thus, we create a simple and effective vaccine design system that uses the worldwide SARS-CoV-2 sequencing data in a meaningful way. While the sequences are publicly available, there is not yet a defined and logical way to make use of them for next-generation vaccines, monitoring monoclonal antibodies, and adding to public-policy measures for preventing spread of emerging lineages”

Reviewers' comments:

Reviewer #1 (Remarks to the Author):

The authors addressed my previous comments. Even though it might take some work, but I will highly recommend including the omicron variant. It would be a good validation if the algorithm correctly predicts the emergence of the omicron variant.

Reviewer #2 (Remarks to the Author):

In the report of DP Maison et al., the authors describe their development of an algorithm for the assessing how evolutionary changes in SARS-CoV-2 correlates with current vaccine and monoclonal antibody efficacies. Their approach is novel in its fusion of two widely used metrics to create a useful tool; they create an algorithm that relies on available prevalence data and Pearson's correlation to identify exponential emergence of key amino acid substitutions and genetic lineages. A practical application of their tool is then demonstrated regarding the status of COVID in Hawai'i, which has a unique racial and ethnic mix living in island communities that are isolated from the US mainland). The results of their analyses can then be studied and expanded to other larger, possibly more complex regions. As a result of their work, a potentially useful tool results for evaluating routes to vaccine and monoclonal antibody production. It would be interesting to see, for example, what the authors would conclude with regard to current vaccines and MABs and current strains of SARS-CoV-2 omicron strains.

The authors have answered my previous questions to my satisfaction. Some minor edits would further improve the manuscript:

(1) L52 - Specify: What type of substitutions; amino acid or nucleotide? Then in L52, the words 'genome evaluation' are used but in L53, 'substitutions' is mentioned without mention if this is in reference to aa or nt. But the test of L53 has to do with spike protein.

(2) L127 - What does 'research' mean? Virology work?

(3) L128 - Certified by CDC? USDA? Other entity?

(4) L144 - do not capitalize 'Isolate'.

L147 and 149 - It is not quite correct to say 'DMEM in FBS'. It is of course FBS in DMEM. Maybe write something like DMEM with 10% FBS...", etc.

L148 - Why say 'wells with Vero cells' and not just 'wells'? Furthermore, better to say that the CELLS were washed. Also, the CELLS, not the cells, were inoculated with virus.

L159 - Why say 'independently'? Is there any reason they would not be done independently?

L175 - "...the third passage.." is an incomplete description. Presumably, the cell growth medium? [ie, Not the cells or mix of cells and cell growth medium].

L281 - Write as 'of concern'.

L301 - Don't capitalize "i" in isolate.

L302 - Best to write as "and resulted in a titer of..."

L303 - Very awkward: Plaque assay quantitated SARS-CoV-2.... Best to write something like "Plaque assays were used to quantitate SARS-CoV-2 isolates, revealing a titer of...for X and ...for Y.

L307 - The genome sequence is deposited in GenBank, not the isolate.

L493 - 494: Language: "vaccines will need to adapt"....Best to write something like "...vaccines will need to be optimized

..."

L519 - Write as 'grow', not as 'grows'.

L529 - Evaluate the strains or sequences?

Reviewer #3 (Remarks to the Author):

The authors have done some commendable work in terms of using existing mutation and correlation analyses that show the distance from the original Wuhan strain and other strains, including South African strains.

There is little novelty though; but the algorithm, no doubt has limitations which includes non-identification of pseudo-codes and if proper demarcations/identification of these are not made, it might even affect the efficiency of vaccine epitope in their in silico vaccine development. One of the attributes of vaccine is its specificity; and one reason why vaccines fail is alteration of immunogenic epitopes, which the authors are aware.

Though, the algorithm can predict deletions and insertion, I advise they consider addressing the deficiencies in their future research.

Reviewer #1 (Remarks to the Author):

Comment 1: The authors addressed my previous comments. Even though it might take some work, but I will highly recommend including the omicron variant. It would be a good validation if the algorithm correctly predicts the emergence of the omicron variant.

Response 1: We thank the reviewer for this comment. The algorithm does predict the emergence of the omicron variant. In the revised version of the manuscript we have included B.1.1.529+BA.* omicron variant in the main manuscript as a figure and the four sublineages (BA.1, BA.1.1, BA.2, and BA.3) as a supplementary figure. Our algorithm predicts B.1.1.529+BA.* as a variant of interest as of 12-02-2021 and variant of concern as of 12-13-2021. The CDC did not classify omicron as a variant of interest and classified it as a variant of concern on 11-29-2021. Omicron was classified for the ~39 amino acid changes in the spike protein. At the time of CDC classification, there were not enough reported omicron whole genome sequences for our algorithm to classify it as emerging. Our algorithm was able to monitor the progress of emergence with standardization. Please note that the algorithm described in this manuscript is a work in progress based on past data and will need to be further tweaked by other researchers to predict future variants for next-generation vaccine design. In summary, the algorithm outlined in this manuscript identified the B.1.1.529 omicron variant two days after CDC defined it as a variant of concern.

Reviewer #2 (Remarks to the Author):

In the report of DP Maison et al., the authors describe their development of an algorithm for the assessing how evolutionary changes in SARS-CoV-2 correlates with current vaccine and monoclonal antibody efficacies. Their approach is novel in its fusion of two widely used metrics to create a useful tool; they create an algorithm that relies on available prevalence data and Pearson's correlation to identify exponential emergence of key amino acid substitutions and genetic lineages. A practical application of their tool is then demonstrated regarding the status of COVID in Hawai'i, which has a unique racial and ethnic mix living in island communities that are isolated from the US mainland). The results of their analyses can then be studied and expanded to other larger, possibly more complex regions. As a result of their work, a potentially useful tool results for evaluating routes to vaccine and monoclonal antibody production. It would be interesting to see, for example, what the authors would conclude with regard to current vaccines and MABs and current strains of SARS-CoV-2 omicron strains.

The authors have answered my previous questions to my satisfaction. Some minor edits would further improve the manuscript:

Comment 1:

(1) L52 - Specify: What type of substitutions; amino acid or nucleotide? Then in L52, the words 'genome evaluation' are used but in L53, 'substitutions' is mentioned without mention if this is in reference to aa or nt. But the test of L53 has to do with spike protein.

Response 1: We thank the reviewer for this comment. We have reworded this sentence to more appropriately distinguish between genomic mutations and amino acid substitutions. The sentence has been changed marked by the highlights in the revised manuscript to:
"The algorithm partitions logarithmic-transformed prevalence data monthly and Pearson's correlation determines exponential emergence of amino acid substitutions (AAS) and lineages. The SARS-CoV-2 genome evaluation indicated 49 mutations, with 44 resulting in AAS."

Comment 2:

(2) L127 - What does 'research' mean? Virology work?

Response 2: We thank the reviewer for this comment.

127: All virus research was conducted in the certified BSL-3 facility at the University of Hawaii John A. Burns School of Medicine Biocontainment Facility.

We have updated this sentence in the revised manuscript as follows:

All SARS-CoV-2 related research was conducted in the University of Hawaii (UH) Institutional Biosafety Committee (IBC) and Environmental Health & Safety Office (EHSO) annually certified BSL-3 facility at the John A. Burns School of Medicine.

Comment 3:

(3) L128 - Certified by CDC? USDA? Other entity?

Response 3: We thank the reviewer for this comment. We have updated this sentence in the revised manuscript as follows:

All SARS-CoV-2 related research was conducted in the University of Hawaii (UH) Institutional Biosafety Committee (IBC) and Environmental Health & Safety Office (EHSO) annually certified BSL-3 facility at the John A. Burns School of Medicine.

Comment 4:

(4) L144 - do not capitalize 'Isolate'.

Response 4: We thank the reviewer for this comment. We have corrected this and all other instances in the revised manuscript.

Comment 5:

L147 and 149 - It is not quite correct to say 'DMEM in FBS'. It is of course FBS in DMEM. Maybe write something like 'DMEM with 10% FBS...', etc.

Response 5: We thank the reviewer for this comment. We have accepted this suggestion and revised the updated manuscript as follows:

Briefly, on the day of the assay, DMEM with 10% FBS was removed from wells, cells were washed twice with serum-free DMEM, and inoculated with multiplicity of infection (MOI) 0.1 and 1, diluted in 500 μ L of DMEM with 2% FBS and incubated at 37°C and 5% CO₂ for two hours.

Comment 6:

L148 - Why say 'wells with Vero cells' and not just 'wells'? Furthermore, better to say that the CELLS were washed. Also, the CELLS, not the cells, were inoculated with virus.

Response 6: We thank the reviewer for this comment. We agree and revised the updated manuscript as follows:

Briefly, on the day of the assay, DMEM with 10% FBS was removed from wells, cells were washed twice with serum-free DMEM, and inoculated with multiplicity of infection (MOI) 0.1 and 1, diluted in 500 μ L of DMEM with 2% FBS and incubated at 37°C and 5% CO₂ for two hours.

Comment 7:

L159 - Why say 'independently'? Is there any reason they would not be done independently?

Response 7: We thank the reviewer for this comment. We agree this word was out of place and have removed it from the revised manuscript.

Comment 8:

L175 - "...the third passage.." is an incomplete description. Presumably, the cell growth medium? [ie, Not the cells or mix of cells and cell growth medium].

Response 8: We thank the reviewer for this comment. Your presumption is correct, viral RNA was extracted from the cell growth medium. The following has been added:

Virus Isolation

Briefly, at the third passage, the cell monolayer and cell supernatant were freeze-thawed three times at -80 C. The supernatant and cell lysate was centrifuged at 5,000 rpm for 15 minutes at 4 C and the supernatant was aliquoted for plaque assay.

The sentence has been revised in the updated manuscript as follows:

Whole Genome Sequencing

For WGS, RNA was extracted from both the VTM and the aliquoted third passage stock virus, as described above, with the QIAamp® Viral RNA Mini Kit (Qiagen, Cat# 52906) following the manufacturer's instructions, as described previously.

Comment 9:

L281 - Write as 'of concern'.

Response 9: We thank the reviewer for this comment. We agree with the reviewer and have updated the sentence in the revised manuscript as follows:

The same algorithm can also be applied for standardizing classifications of SARS-CoV-2 lineages as of interest or of concern.

Comment 10:

L301 - Don't capitalize "i" in isolate.

Response 10: We thank the reviewer for this comment. We have corrected this and all other instances in the revised manuscript.

Comment 11:

L302 - Best to write as "and resulted in a titer of..."

Response 11: We thank the reviewer for this comment. This suggestion has been incorporated into the revised manuscript as follows:

A stock of the SARS-CoV-2, isolate USA-HI498 2020 was produced following three passages in Vero E6 cells.

Comment 12:

L303 - Very awkward: Plaque assay quantitated SARS-CoV-2.... Best to write something like "Plaque assays were used to quantitate SARS-CoV-2 isolates, revealing a titer of...for X and ...for Y.

Response 12: We thank the reviewer for this comment. We agree this is an improvement and have incorporated this comment into the revised manuscript as follows:

Plaque assays were used to quantitate SARS-CoV-2 isolates. The virus stocks were titered at 1.28×10^7 PFU/mL for SARS-CoV-2 isolate USA-HI498 and 3.88×10^7 PFU/mL for SARS-CoV-2 USA-WA1/2020.

Comment 13:

L307 - The genome sequence is deposited in GenBank, not the isolate.

Response 13: We thank the reviewer for this comment. Yes, the genome sequence is deposited in GenBank and the corresponding isolate is deposited in BEI Resources. We have revised the sentence in the updated manuscript to improve clarity.

SARS-CoV-2, isolate USA-HI498 2020 is deposited in BEI Resources Cat# NR-56130 and the corresponding whole genome sequence in the GenBank (Accession Number MZ664037).

Comment 14:

L493 - 494: Language: "vaccines will need to adapt"....Best to write something like "..vaccines will need to be optimized..."

Response 14: We thank the reviewer for this comment. We have updated the sentence to incorporate the suggestion in the revised manuscript as follows:

As S1 is shed in the coronavirus model of fusion,⁷² and S2 is responsible for fusion, the diversity of S1 in SARS-CoV-2 and the epitope targeting concentration in S1, indicate that SARS-CoV-2 vaccines will need to be optimized.

Comment 15:

L519 - Write as 'grow', not as 'grows'.

Response 15: We thank the reviewer for this comment. We have corrected the sentence in the revised manuscript according to this suggestion.

Comment 16:

L529 - Evaluate the strains or sequences?

Response 16: We thank the reviewer for this comment. We have revised the updated manuscript to reflect that we evaluate both the strains and the sequences.

Here, we isolate and characterize SARS-CoV-2 in cell culture and analyze the whole-genome sequences of these isolates and sequences deposited in GISAID to develop an algorithm for next-generation vaccine design.

Reviewer #3 (Remarks to the Author):

Comment 1:

The authors have done some commendable work in terms of using existing mutation and correlation analyses that show the distance from the original Wuhan strain and other strains, including South African strains.

There is little novelty though; but the algorithm, no doubt has limitations which includes non-identification of pseudo-codes and if proper demarcations/identification of these are not made, it might even affect the efficiency of vaccine epitope in their in silico vaccine development. One of the attributes of vaccine is its specificity; and one reason why vaccines fail is alteration of immunogenic epitopes, which the authors are aware.

Though, the algorithm can predict deletions and insertion, I advise they consider addressing the deficiencies in their future research.

Response 1: We thank the reviewer for this comment and value the reviewers critical insight of our data. The algorithm described in this manuscript is a work in progress and will need to be further modified to predict future variants.